# Non-orthogonal optical multiplexing empowered by deep learning

Tuqiang Pan [1,2,3], Jianwei Ye[1,2,3], Haotian Liu[1,2], Fan Zhang[1,2], Pengbai Xu[1,2], Ou Xu[1,2], Yi Xu [1,2] ✉ & Yuwen Qin [1,2] ✉

Orthogonality among channels is a canonical basis for optical multiplexing featured with division multiplexing, which substantially reduce the complexity of signal post-processing in demultiplexing. However, it inevitably imposes an upper limit of capacity for multiplexing. Herein, we report on non-orthogonal optical multiplexing over a multimode fiber (MMF) leveraged by a deep neural network, termed speckle light field retrieval network (SLRnet), where it can learn the complicated mapping relation between multiple non-orthogonal input light field encoded with information and their corresponding single intensity output. As a proof-of-principle experimental demonstration, it is shown that the SLRnet can effectively solve the ill-posed problem of non-orthogonal optical multiplexing over an MMF, where multiple non-orthogonal input signals mediated by the same polarization, wavelength and spatial position can be explicitly retrieved utilizing a single-shot speckle output with fidelity as high as ~ 98%. Our results resemble an important step for harnessing non-orthogonal channels for high capacity optical multiplexing.

Multiplexing is a cornerstone for optical communication, where physical orthogonality among multiplexing channels is a prerequisite for massively-encoded transmission of information[1,2]. For example, division multiplexing becomes a canonical form for increasing the capacity of fiber communication, such as space division multiplexing[1,3,4], wavelength division multiplexing[1,5], polarization division multiplexing[6,7] and mode division multiplexing[5,8]. However, the division nature and the orthogonal paradigm of these multiplexing mechanisms inevitably impose an upper limit of multiplexing capacity[1-10]. If the orthogonal paradigm of optical multiplexing can be broken, it could be a step forward for realizing non-orthogonal optical multiplexing, which will become a promising way to meet the challenge of information capacity crunch. Considering the demultiplexing of multiple orthogonal signals, the transmission matrix method[11-15] can tackle this issue even over a strongly scattering medium, such an MMF. While non-orthogonal optical multiplexing over an MMF can be referred to multiplexing input channels possessing non-orthogonal

polarizations, the same wavelength, and the same spatial position, where their polarizations are even the same for the typical non-orthogonal scenario. In this case, the inverse transmission matrix method fails to decode the multiplexing signals with the same polarization and wavelength using a single-shot intensity detection, as schematically shown in Fig. 1a.

Recently, deep learning has been widely used in optics and photonics for inverse design of optical devices[16,17] and computational optics[18-21]. Specifically, deep neural network has been utilized to improve the performance of orthogonal multiplexing over a multiple scattering medium[22-33]. To date, however, all the reported multiplexing scenarios strictly rely on the physical orthogonality among multiplexing channels[11-15,22-33]. There is no attempt to leverage the nonlinear modelling capability of deep learning to achieve the non-orthogonal optical multiplexing over an MMF, resembling an alluring but still open question. Unfortunately, even multiplexing of non-orthogonal channels mediated by the same polarization or wavelength in a single mode

[1]Key Laboratory of Photonic Technology for Integrated Sensing and Communication, Ministry of Education, Guangzhou 510006, China. [2]Guangdong Provincial Key Laboratory of Information Photonics Technology, Institute of Advanced Photonic Technology, School of Information Engineering, Guangdong University of Technology, Guangzhou 510006, China. [3]These authors contributed equally: Tuqiang Pan, Jianwei Ye. ✉e-mail: yixu@gdut.edu.cn; qinyw@gdut.edu.cn

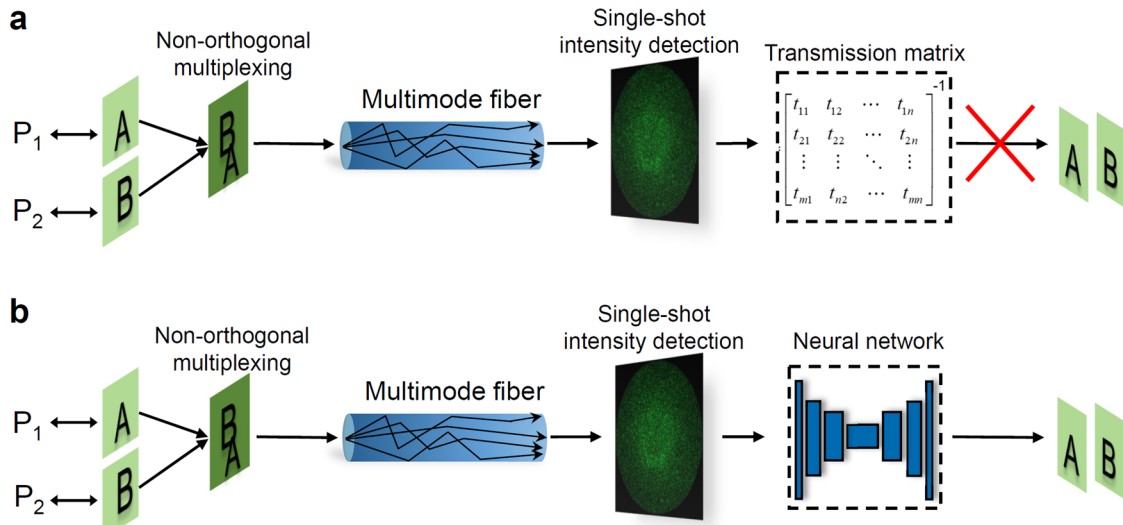

**Fig. 1 | Schematic of non-orthogonal optical multiplexing over an MMF. a** The non-orthogonal multiplexing information cannot be retrieved by the inverse transmission matrix method using a single-shot intensity detection. **b** Schematic of deep learning-based non-orthogonal multiplexing under multiple scattering of an MMF. As long as the neural network is well trained, the information of each channel can be retrieved utilizing the single-shot intensity output.

fiber remains very challenging, which is due to the lack of effective demultiplexing method or overburdened digital signal processing[1]. Therefore, developing a new methodology for decoding information encoded in non-orthogonal input channels is of vital importance for the ultimate optical multiplexing.

In this work, we show that preliminary non-orthogonal optical multiplexing through an MMF can be achieved empowered by the SLRnet. As a proof-of-concept demonstration, multiplexing transmission of information through an MMF, including general natural scene images, uncorrelated random binary data and images not belong to the same type of training dataset, can be realized utilizing non-orthogonal input channels, as schematically shown in Fig. 1b, facilitating the realization of non-orthogonal multiplexing transmission of optical information. Building a complicated relationship between the non-orthogonal input channels and the output through the data-driven technology, a well-trained deep neural network can retrieve the encoded information of the non-orthogonal channels merely using a single-shot output intensity. Even non-orthogonal multiplexing channels sharing the same polarization, wavelength and input spatial region can be effectively decoded. It is anticipated that our results would not only stimulate various potential applications in optics and photonics, but also inspire explorations in more broader disciplines of information science and technology.

## Results

### Principle

The single channel input-output relationship of an MMF can be described by a transmission matrix, as shown by the following equation:

$$\vec{E}_{out} = T\vec{E}_{in} \tag{1}$$

Here, $T$ is the transmission matrix for a multiplexing channel through the MMF with a given input polarization and $\vec{E}_{out}$ and $\vec{E}_{in}$ are the output and input light fields, respectively. Notably, $\vec{E}_{in/out}$ is a complex number that contains both amplitude and phase information encoded in space (i.e. $X$ and $Y$ dimensions), which can be expressed as:

$$\vec{E}_{in} = \mathbf{a}A(x,y)e^{j\varphi(x,y)} \tag{2}$$

In this equation, $A(x,y)$ and $\varphi(x,y)$ represent the amplitude and phase distributions of the incident light field, respectively. $\mathbf{a}$ indicates the unit vector of electric field. If there is only an input field, the

transmission matrix can be calibrated and the scrambled input wavefront can be retrieved[11,14,15].

If the incident wavefront is a superposition of multiple non-orthogonal light fields, it becomes:

$$\vec{E}_{in} = \sum_{i=1}^{n} \vec{E}_i = \sum_{i=1}^{n} \mathbf{a}_i A_i(x,y)e^{j\varphi_i(x,y)} \tag{3}$$

where $\vec{E}_i$ is the $i_{th}$ incident light field. As shown in Fig. 1a, multiple amplitude and phase encoded wavefronts with the same polarization states are superimposed and coupled to the proximal end of the MMF, resulting in a single speckle output at the distal end of the MMF. The output speckle intensity recorded by the CMOS camera can be expressed as follows:

$$I = |\vec{E}_{out}|^2 = |\sum_{i=1}^{n} T_i \vec{E}_i|^2 = H(C_n), C_n = \{(A_i,\varphi_i); i = 1,2,...,n\} \tag{4}$$

where $H(\cdot)$ represents the mapping relationship between multiple input light fields $C_n$ and the single output speckle $I$ of the MMF. Here, $C_n$ indicates $n$ combinations of information encoded input amplitude and phase. It should be pointed out that the transmission matrix $T_i$ is different even for the multiplexing channels with parallel polarization because of the residual optical asymmetry during multiplexing and coupling to the MMF, such as slightly different k-vectors of the multiplexed beams. According to Eq. (4), the inverse transmission matrix method cannot retrieve each $\vec{E}_i$ using a single-shot intensity detection, as shown in Fig. 1a. In order to realize demultiplexing of non-orthogonal signals through the MMF from a single-shot output speckle, the inverse mapping $H^{-1}$ of the above equation

$$H^{-1}(I) = C_n \tag{5}$$

should be obtained. There is no physics-based theory reported so far, which can effectively obtain $H^{-1}$ when $n > 1$.

In this case, data-driven deep learning methods become an effective means to solve this problem where multiple input $\vec{E}_i$ are non-orthogonal light fields with non-orthogonal polarizations, the same wavelength, and the same spatial position. A typical supervised learning method relies on a sequence of labelled data, $(C_{nk}, I_k), k = 1, 2, \ldots, K$, to obtain the mapping function $R$ by learning

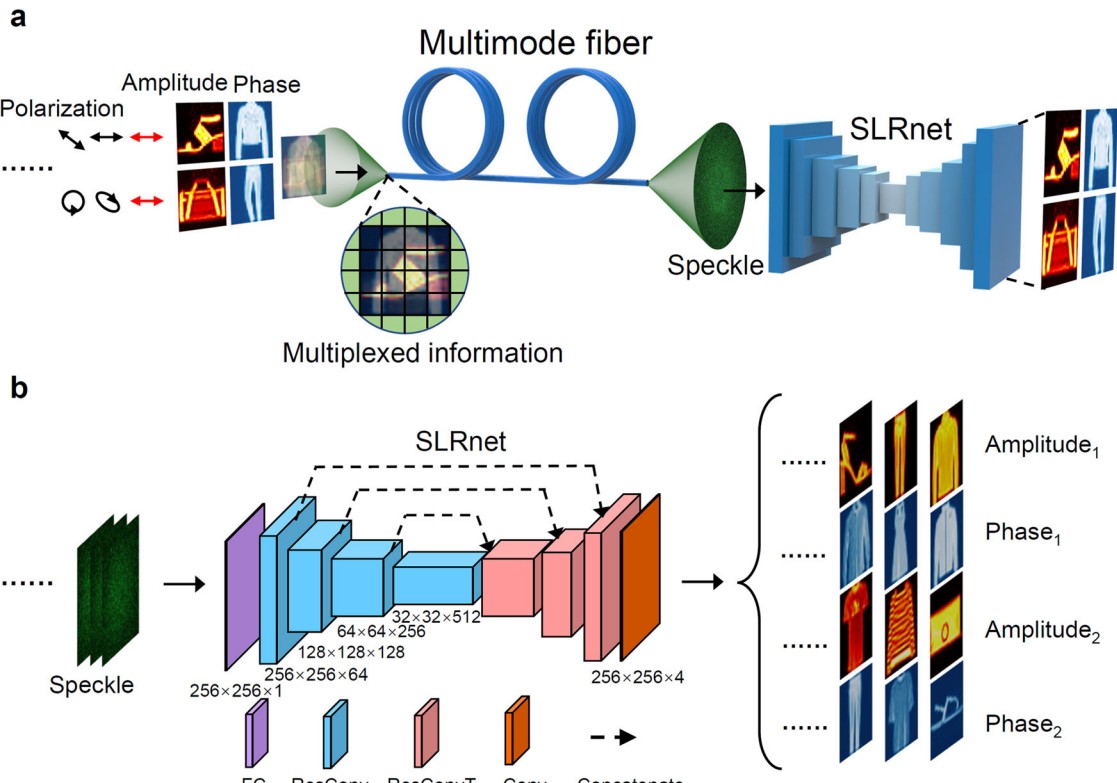

**Fig. 2 | The non-orthogonal optical multiplexing over an MMF enabled by deep learning. a** Non-orthogonal multiplexed information encoded in the amplitude, phase and polarization dimensions are superimposed at the proximal end of the MMF, resulting in a speckle output at the distal end of the MMF. Then the encoded multidimensional information can be unambiguously retrieved from a single-shot output speckle utilizing the SLRnet. The polarization states are outlined and the same wavelength is used. The grids superimposed on the input information indicate the information units in both amplitude and phase dimensions. **b** The architecture of SLRnet is composed of a fully connected (FC) block, four residual convolutional blocks with down sampling, three residual transposed convolutional blocks with up sampling, and one output convolutional layer for channel compressing. Skip connections are established among the first three down-sampling and the up-sampling modules. The sizes of feature map for each block are marked in the insets. ResConv Residual convolutional block, ResConvT Residual transposed convolutional block, Conv Convolutional layer. All images are adopted from the Fashion-MNIST dataset[41].

from the training set $Q = \{(C_{nk}, I_k), k = 1, 2, \ldots, K\}$. It should be emphasized that the information encoded in non-orthogonal multiplexing channels is effectively orthogonal in the labelling of $C_{nk}$. At the same time, the residual asymmetry of the optical paths plays an important role in the non-orthogonal optical multiplexing through the MMF. The residual asymmetry will be leveraged by the multiple scattering of MMF, which can facilitate the solution of multiple-to-one mapping relationship. As a result, the following equation:

$$R_{\theta'} = \underset{\theta \in \Theta}{\arg\min} |R_{\theta}(I_k) - C_{nk}|, \forall (C_{nk}, I_k) \in Q \qquad (6)$$

is optimized, where $R_{\theta}$ is the mapping function determined by the weight of deep learning network $\theta \in \Theta$ and $\Theta$ is all possible weight parameters of the network. The well trained $R_{\theta'}$ can retrieve the information encoded in input amplitude and phase $\widetilde{C_n}$ utilizing the output speckle $I$ not belonging to $Q$, i.e., $\widetilde{C_n} = R_{\theta'}(I)$. This neural network builds an approximate relationship that maps the speckle intensity at the distal end of the MMF to the distributions of amplitude and phase for several input light fields at the proximal end of the MMF, where the training of the network relies on the dataset using pairs of output speckles and their corresponding input wavefronts. In other words, the multiplexed non-orthogonal input light fields can be demultiplexed by:

$$H^{-1}(I) \approx R_{\theta'}(I) = \widetilde{C_n} \qquad (7)$$

## Neural network architecture

According to the principle analysed above, deep neural network is capable of retrieving non-orthogonal optical multiplexing signals from a single speckle output of the MMF. As shown in Fig. 2a, multiple amplitude and phase encoded information mediated by arbitrary combinations of polarizations can be effectively retrieved by the SLRnet after propagating in the MMF. Even the typical scenario of non-orthogonal input channels with the same polarization, wavelength and input spatial region can be explicitly decoded. This is enabled by a deep neural network whose architecture is shown in Fig. 2b, which is a variant of Unet according to the unique multiple scattering process of the MMF. It consists of a fully connected (FC) layer and a ResUnet[34], whose main advantages over Unet are as follows: (1) a FC layer is introduced before the input of Unet to enhance the fitting and generalization ability of the network. The introduction of the FC layer can effectively undo the nonlocal dispersion of the MMF, which improves the performance of demultiplexing multidimensional encoded information using a single speckle output. The ResUnet is used for denoising and post-processing the multiplexing information towards the ground truth, which is similar to the convnet proposed recently[25]. In addition, the convolutional layer can also facilitate the manipulation of multichannel outputs in the non-orthogonal multiplexing without increasing the training burden; (2) a large number of skip connections are introduced in the encoder-decoder path to enhance the degeneration-free propagation of data in the network (See "Methods" section for details). To facilitate the experimental verification, $n$ is chosen as 2, where the non-orthogonal inputs contain two light field

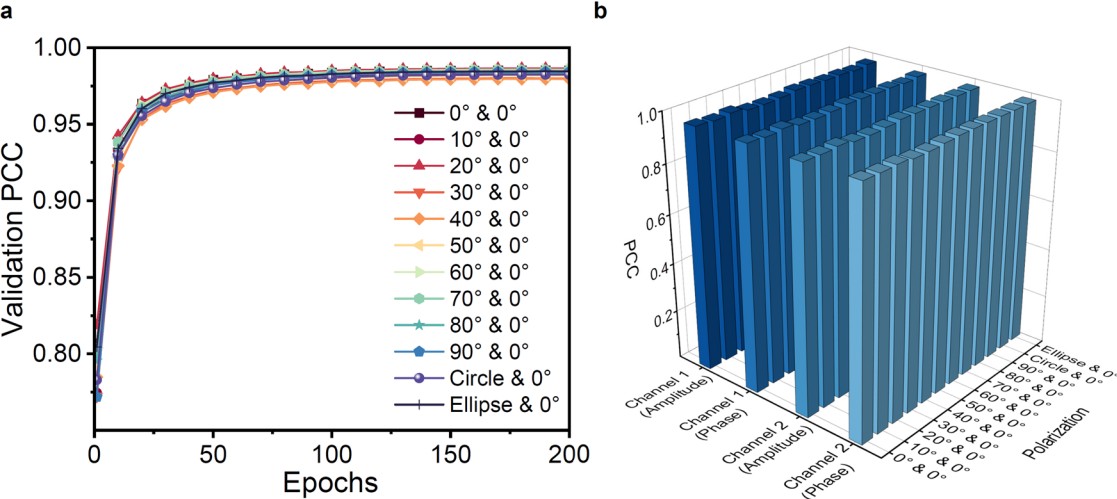

**Fig. 3 | Performance of the non-orthogonal multiplexing using the SLRnet.**
**a** Averaged PCCs of the validation dataset during training procedures, where multiplexing scenarios of two input channels over a 1 m MMF with arbitrary polarization combinations are shown. The angles of polarizations with respect to the horizon line are indicated. Here, circle and ellipse indicate circular and elliptical polarizations, respectively. **b** The PCCs of retrieved information in different multiplexing channels at the final epoch. PCC Pearson correlation coefficient.

channels mediated by arbitrary polarization combinations. Each light field channel is composed of spatially encoded information in both amplitude $A(x, y)$ and phase $\varphi(x, y)$ dimensions, respectively, resulting totally four multiplexing channels for transmitting independent information. Supervised learning is applied during the training process of the network. The speckle at the output of the MMF is used as the input of the SLRnet, where the network outputs predicted four matrices containing both the encoded amplitude and phase information in two light field channels. It means that the encoded information in non-orthogonal multiplexing channels is orthogonally labelled, when the network is trained. This is the key point for decoding information encoded in non-orthogonal channels.

**Experimental results**

The case when the length of MMF is 1 m is considered first. Figure 3a presents the evolution of retrieved fidelity for two multiplexed light field channels with arbitrary combinations of polarization states during the training process of SLRnet. In total, there will be four encoded channels in the amplitude and phase dimensions, where they can be non-orthogonal depending on the polarization states. Here, the retrieved fidelity is measured by Pearson correlation coefficient (PCC). Twelve different combinations of polarization states, including linear, circular and elliptical polarizations, are considered. In these cases, their wavelengths are the same (See "Methods" section for details). As can be seen from this figure, the evolutions of the retrieved PCC utilizing the same training configuration of SLRnet are larger than 0.97 after 100 epochs, indicating the condition of Eq. (7) is approached, where $H^{-1}(I) \approx R_{\theta'}(I)$. At the same time, the evolutions of retrieved fidelity for twelve multiplexed scenarios are basically the same, which showcases excellent robustness of non-orthogonal multiplexing with respect to arbitrary polarization combinations. In particular, the case of 0° & 0° demonstrates the successful multiplexing using channels with the same polarization, wavelength and input spatial region, validating the promising capability of non-orthogonal optical multiplexing. Furthermore, Fig. 3b provides the retrieved fidelity in each amplitude and phase multiplexing channel using different combinations of polarizations, respectively. The averaged retrieved fidelity in the amplitude and phase dimensions are almost the same (~0.98), which highlights the capability of SLRnet in demultiplexing information encoded in multiple non-orthogonal input channels (see Supplementary Note 1 for the results measured by structure similarity index measure (SSIM)).

To provide a sensory evaluation of the retrieved information encoded in the wavefront, typical demultiplexing results for four polarization combinations (0° & 0°, 0° & 10°, 0° & 90°, and 0° & Ellipse) are presented in Fig. 4. The corresponding input polarization states are outlined in the figure. And the retrieved fidelity measured by SSIM and PCC for all these cases are also provided, respectively. As can be seen from these results, four grayscale images multiplexed in the amplitude and phase of the input wavefronts using the same polarization can be effectively demultiplexed utilizing a single-shot speckle output. The retrieved fidelity of other results under different combinations of polarizations are similar, indicating the SLRnet enables the unprecedented multiplexing of non-orthogonal input channels even when the encoded wavefronts are scrambled by the MMF. To further consolidate the superiority of SLRnet in a more realistic scenario, the non-orthogonal optical multiplexing results using the same polarization state over a 50 m MMF are presented (see "Methods" section for details), as shown in Fig. 5. As can be seen in Figs. 4 and 5, the demultiplexing results of the 1 m MMF is better than the 50 m case. This is because the scattering properties of a longer MMF are much easier to be affected by the environment. The demultiplexing performance can be further improved by optimizing the network architecture. The high fidelity achieved for multiplexing non-orthogonal channels utilizing the same polarization, wavelength and input spatial position indicates that the SLRnet is an effective means for multiplexing non-orthogonal channels in an MMF.

In order to showcase the generality of the SLRnet for a diverse set of images, various experimental results considering more complicated grayscale encoded information from the CelebA face dataset[35], random binary data whose digital information is uncorrelated, general natural scene images from the ImageNet database[36], and snapshots in Muybridge recordings not belong to the same type of training dataset are presented (see Supplementary Note 2). Typical results for general natural scene images are shown in Fig. 6a, where the achieved averaged SSIM/PCC is 0.737/0.905. To further increase the modulation precision of the wavefront, the information is only encoded in the phase dimension of two non-orthogonal channels. The achieved averaged fidelity is 0.819/0.945 (SSIM/PCC) as shown in Fig. 6b, which is substantially improved compared with the complex modulation case. At the same time, the achieved typical fidelity for images not belong to the ImageNet database can be up to 0.907/0.986 (SSIM/PCC), as shown in Fig. 6c, indicating the good generalization of the

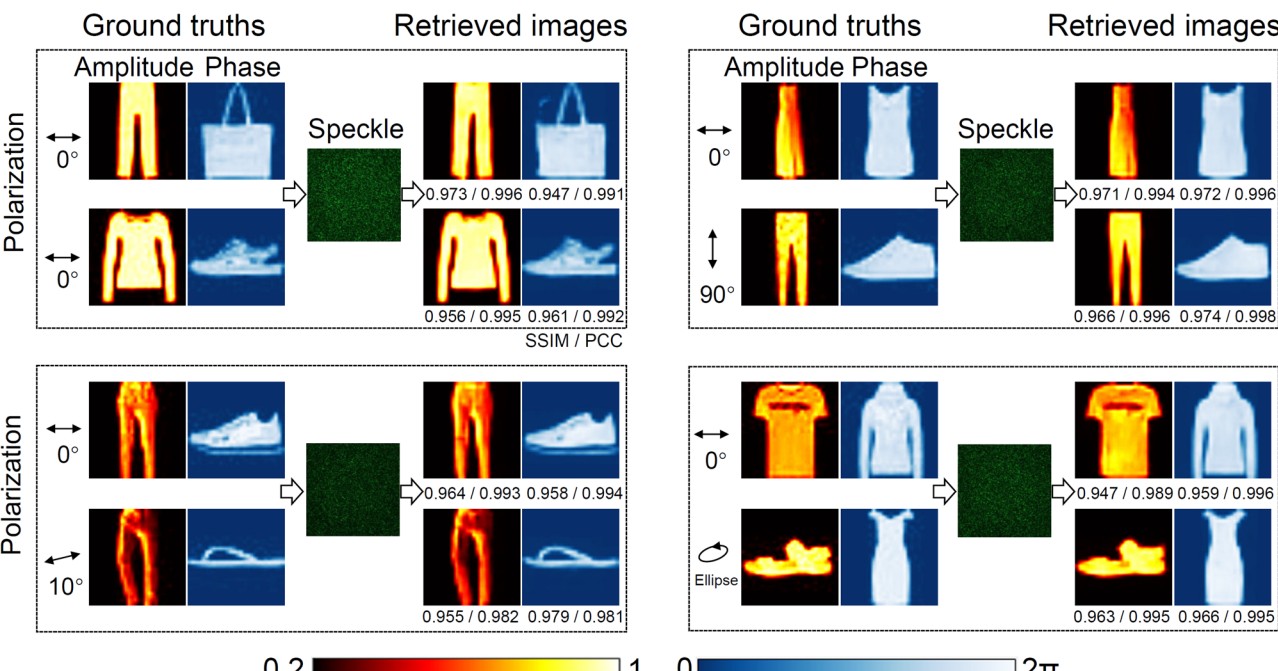

**Fig. 4 | Results of non-orthogonal multiplexing over a 1 m MMF.** The ground truths, the speckle output and the corresponding retrieved light field information by the SLRnet using a single-shot speckle output are shown, where their corresponding SSIM and PCC are given, respectively. Colorbars are also provided for the grayscale images encoded in the amplitude and phase. These images are adopted from the Fashion-MNIST dataset[41].

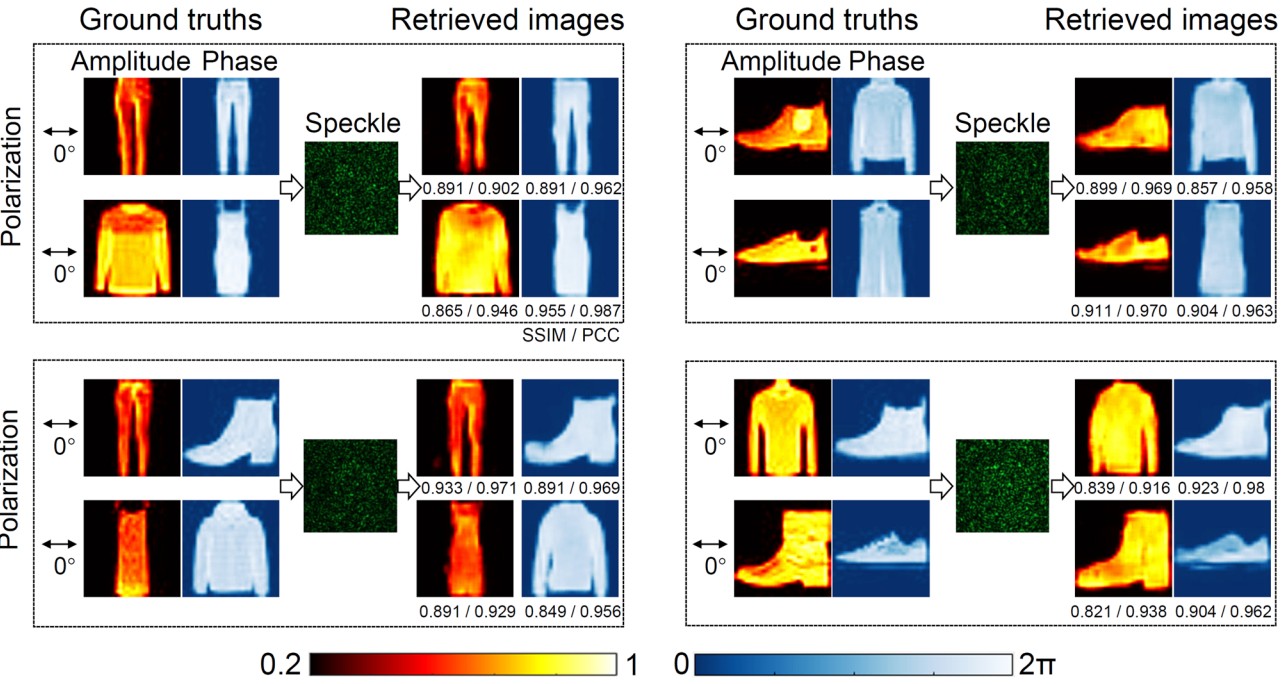

**Fig. 5 | Results of non-orthogonal multiplexing over a 50 m MMF.** The ground truths, the speckle output and the corresponding retrieved light field information by the SLRnet using a single-shot speckle output are shown, where their corresponding SSIM and PCC are given, respectively. Colorbars are also provided for the grayscale images encoded in the amplitude and phase dimensions. These images are adopted from the Fashion-MNIST dataset[41].

SLRnet. All these results further validate that the SLRnet has an excellent ability to retrieve multiplexed information encoded in the non-orthogonal input channels.

## Discussion

We demonstrate a concept of non-orthogonal optical multiplexing over an MMF empowered by deep learning utilizing the SLRnet. Up to five optical degrees of freedom with non-orthogonal combinations of amplitude, phase, polarization and two-dimensional space (*X* and *Y*) are utilized for the non-orthogonal multiplexing, where the multiplexed information in the proximal end of MMF can be effectively demultiplexed using a single-shot speckle output at the distal end of MMF. The experimental results reveal that the proposed SLRnet can achieve high-fidelity (~98%) retrieval of multidimensional light field

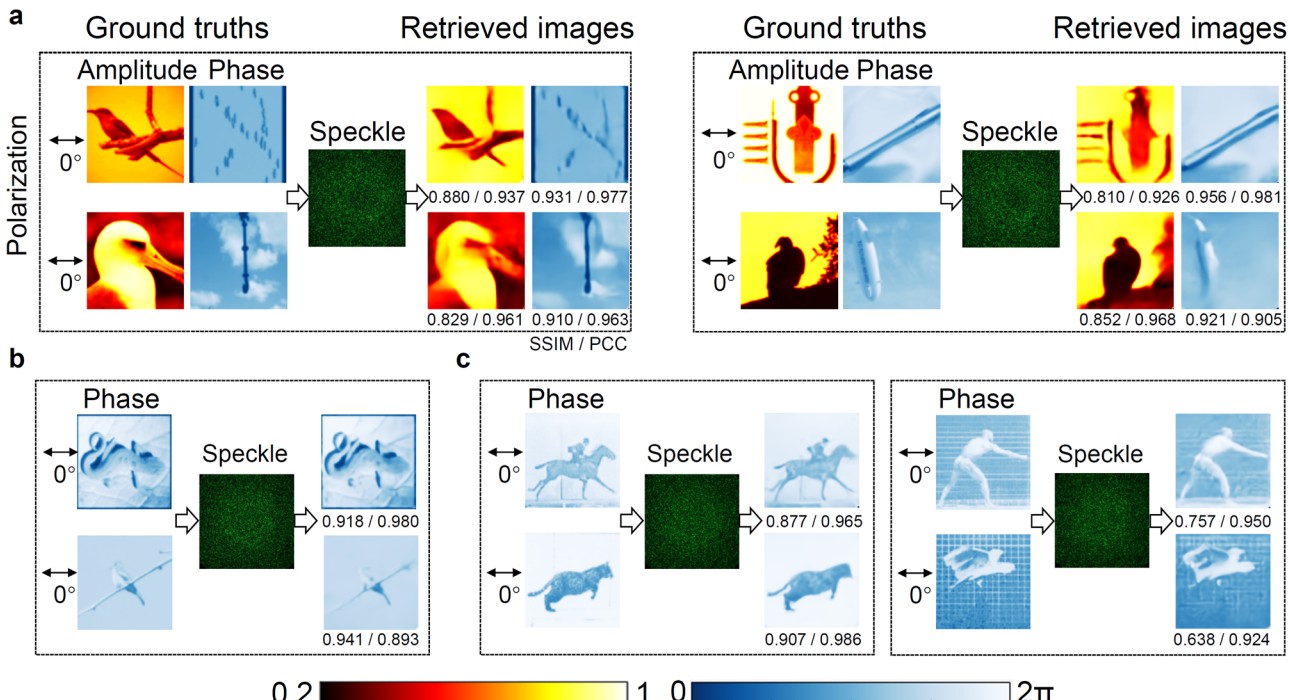

**Fig. 6 | Results of non-orthogonal multiplexing of general natural scene images and images not belong to the ImageNet database over a 1 m MMF. a** The ground truths, the speckle outputs, and the corresponding retrieved light field information by the SLRnet are shown, where their corresponding SSIM and PCC are given. **b** The corresponding results for the non-orthogonal multiplexing of phase encoded information. The images in (**a**) and (**b**) are from the ImageNet database[36]. **c** The results for the non-orthogonal multiplexing of phase encoded information using snapshots of Muybridge recordings from the 1870s that marked the historically important breakthrough of the first ever high-speed photography images. These images are not belong to the ImageNet database used for the training of the neural network.

transmitted over an MMF. The performance of SLRnet is comparable to or even exceeding the reported results of orthogonal optical multiplexing in optical degrees of freedom, fidelity and spatial channel numbers (see Supplementary Note 3 for more details). At the same time, both the training and validation datasets contain the influence from the environment (see Supplementary Note 4 for more details). According to the retrieved results demonstrated above, the trained SLRnet possesses certain robustness against the perturbation from the environment. More robust demultiplexing can be achieved by using joint training of data collected at different environments[27,29]. If more than two input channels are involved in the non-orthogonal optical multiplexing, the total amount of data should be increased for achieving similar fidelity.

Although the proposed concept of non-orthogonal optical multiplexing over an MMF cannot be directly used in medical diagnosis at this stage, which generally requires unity fidelity, the non-orthogonal multiplexing of uncorrelated binary digital information with high accuracy indicates a step forward for realizing non-orthogonal multiplexing transmission of optical information through an MMF. It is anticipated that our results could not only pave the way for harnessing the high throughput MMFs for communication and information processing, but also might provide a paradigm shift for optical multiplexing in optics and beyond, which can substantially improve the degrees of freedom and capacity of optical systems.

Furthermore, light has many physical quantities that can be used to encode information. It is also anticipated that more optical degrees of freedom can be used for non-orthogonal optical multiplexing, such as the wavelengths and orbital angular momentum. There is room for optimizing the performance of deep neural network, where the achieved fidelity, efficiency and generalization should be further improved. Recent studies have shown that transformer structures based on self-attention mechanism may achieve higher fidelity. And the network based on the prior Fourier transform can result in superior external generalization[37]. There are still challenges to overcome in this data-driven approach. A typical one is the ability to multiplex information with higher capacity will require exponentially increasing amounts of data (see Supplementary Note 2). Adding a physically-informed model of the MMF system in the deep neural network could be an effective solution for this challenge, which would also boosting the demultiplexing fidelity[24,25]. In addition, incorporating transfer learning could substantially reduce the amount of data required for training.

## Methods
### Experimental setup
A monochromatic laser with a power of 50 mW ($\lambda$ = 532 nm, MSL-S-532 CH80136, CNI) is used as the light source (see Supplementary Note 5 for more details regarding to the experimental setup), where it can be generalized to other wavelengths. The horizontal polarized laser beam is collimated and expanded by an objective lens ($Obj_1$) and a lens ($L_1$). Then it is divided into two beams of the same size using a dual-channel diaphragm. A phase-only spatial light modulator (SLM, PLUTO-NIR, Holoeye) is used for realizing amplitude and phase modulations simultaneously, which will be elaborated in the following. $L_2$, iris and $L_3$ constitute a 4f filtering system, and the first-order diffracted light is selected at the focal plane to obtain the targeted amplitude and phase encoded light field. A wave plate is used to adjust the polarization state for one of the laser beams while the other beam keeps the horizontal polarization state unchanged. They are coherently superimposed with a non-polarized beam splitter cube (NPBS), forming two collinear beams with arbitrary polarization combinations. Then, two multiplexed laser beams are coupled into an MMF by an objective lens ($Obj_2$). And the outgoing light field from the MMF is collected by another objective lens ($Obj_3$), where the output speckle is recorded by

a CMOS camera (MER-231-41U3C-L, Daheng Imaging). The captured speckles are translated to grayscale images. Two kinds of MMFs are tested: one is 1 m (Newport, diameter $\phi = 400\,\mu$m, NA = 0.22) while the other one is 50 m (YOFC, diameter $\phi = 105\,\mu$m, NA = 0.22). Both MMFs are step-index MMFs.

### Data acquisition and preprocession
The parameters of all used datasets and their corresponding averaged fidelity are summarized in Supplementary Note 6. Each dataset is divided into a training set (90%) and a validation set (10%). The data in the validation set is uniformly sampled in its corresponding dataset. The resolution of the speckle output fed to the network is 200 × 200. All the images encoded in the amplitude dimension are scaled from 0-1 to 0.2-1.

### Amplitude and phase modulation scheme
To achieve simultaneous phase and amplitude modulations of the light field by a phase-only SLM, a complex amplitude modulation algorithm based on phase-only hologram coding is used[38]. The amplitude information is encoded into the phase information by modifying the spatial diffraction efficiency, where the target light field information is obtained by filtering.

### Network structures
The proposed SLRnet consists of a FC layer and a ResUnet. In the FC block, a linear layer and an adaptive average pooling layer are used to control the size of the layer's output. The linear layer can increase fitting and generalization abilities of the network. At the same time, the ResUnet introduces abundant skip connections to the Unet structure and accelerates the training process of network[34]. In this case, the ResUnet consists of four parts, including residual convolutional blocks (ResConv), residual transposed convolutional blocks (ResConvT), an output convolutional layer (Conv), and skip connections, as shown in Fig. 2b. The ResConv achieves downsampling feature extraction by using a convolutional layer with a stride of 2, while ResConvT achieves upsampling reconstruction by using a transposed convolutional layer with a stride of 2. And there are skip connections between every two symmetrically arranged ResConv and ResConvT for concatenating channels. Finally, the channel matching output is carried out through the Conv with the convolution kernel size of 1 × 1. After adding batch normalization to the convolution layer of ResConv and ResConvT, the convergence of network training is faster. At the same time, adding a Rectified Linear Unit (ReLU) activation function introduces nonlinear factors to enhance the fitting ability of the network (see Supplementary Note 7 for detailed structure).

### Training configuration
The SLRnet is implemented using python 3.9.13 in PyTorch 1.13.0. It is trained with the AdamW optimizer[39], an improved version of the Adam optimizer with better generalization performance. The initial learning rate is set at $2 \times 10^{-4}$, rising to $1 \times 10^{-3}$ after five epochs of warm-up, and subsequently dropping to 0 at the last epoch according to the cosine annealing schedule (see Supplementary Note 8 for the learning rate curve). This is an advanced learning rate adjustment strategy[40], where its loss function value during the training process does not fluctuate significantly. The total training epoch is set to 200 to ensure that the training is converged. Mean absolute error is selected as the loss function to train the network.

### Reporting summary
Further information on research design is available in the Nature Portfolio Reporting Summary linked to this article.

## Data availability
The example dataset for the non-orthogonal multiplexing of phase encoded information is available at: https://doi.org/10.5281/zenodo. 10391031. Any additional data are available from Yi Xu (yixu@gdut.edu.cn) upon request. Source data are provided with this paper.

## Code availability
The Python codes used in this paper are available at https://doi.org/10.5281/zenodo.10391031.

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

## Acknowledgements

The authors would like to thank Cheng-Wei Qiu, Songnian Fu, Jianping Li and Meng Xiang for their inspired suggestions and comments. This work was supported by National Key R&D Program of China under grant no. 2018YFB1801001 (Y.Q.), National Natural Science Foundation of China under grant nos. 62222505 (Y.X.) and 62335005 (Y.X.) and Guangdong Introducing Innovative, Entrepreneurial Teams of "The Pearl River Talent Recruitment Program" under grant nos. 2019ZT08X340 (Y.Q.) and 2021ZT09X044 (Y.X.).

## Author contributions

Y.X. conceived the idea. T.P., J.Y. and H.L. conducted the experiments with the assistance of F.Z., P.X. and O.X. T.P. designed and trained the deep learning network. Y.X. and T.P. analysed the results. Y.X. and T.P. wrote the manuscript with inputs from all authors. All authors reviewed the manuscript. Y.X. and Y.Q. supervised this project.

## Competing interests

The authors declare no competing interests.
