## [Peer Review File · Nature Communications]

REVIEWER COMMENTS

Reviewer #1 (Remarks to the Author):

This paper takes a task which has been approached by a number of previous papers (cited in the work), and adds a novel task complication of non-orthogonal multiplexed images and a new combination of deep learning layers which you apply to the standard approximate inversion problem from a speckle image.

The paper poses the task on p1 "If the orthogonal paradigm of optical multiplexing can be break, unprecedented non-orthogonal multiplexing can be achieved, which will become a promising way to meet the challenge of information capacity crunch." then making the very strong claim "In this work, we show that non-orthogonal optical multiplexing can be explicitly achieved empowered by the SLRnet. As a proof-of-concept experiment, we demonstrate that multiplexing transmission of optical information in a MMF can be realized utilizing non-orthogonal channels".

Unfortunately, although there is valuable, interesting empirical work in the paper, which could inspire other researchers, I do not believe that the paper justifies such a strong claim, and therefore cannot be published in its current form.

The initial concern builds upon the author's own observations on p3 "In order to realize demultiplexing of non-orthogonal signals through the MMF from a single-shot output speckle, the inverse mapping H^{-1} of the above equation

$$H^{-1}(I) = Cn \quad (5)$$

should be obtained. There is no physics-based theory reported so far that can effectively obtain H^{-1} when $n > 1$."

If we cannot effectively find the solution to this clearly ill-posed inverse problem, we need to clearly discuss the assumptions involved in the approximation provided by the neural network. We also need to think carefully about the nature of empirical evidence, and the choice of data to demonstrate 'multiplexing transmission of optical information in a MMF' in a convincing manner. What sort of images can this system actually communicate?

The dataset chosen fMNIST is a commonly used dataset as an introductory tool in machine learning, but is well-known to be a very simple task, with 10 major classes, and many examples in each class which are

very similar to each other. It is also an easy task to create autoencoders which can reproduce the images, and many images in each class are very similar to each other. E.g. a simple UMAP low dimensional embedding can take fMNIST data and generate 10 separate classes in an unsupervised fashion, and can still do this if the images are transformed by random unitary matrices, akin to a MMF transformation). (a fully-connected network would do as well with the raw images as with the unitarily transformed ones. A conv-net would not do as well with the transformed data, given its spatial assumptions, but on fMNIST even a simple fully-connected network gets over 96% accuracy, which highlights how limited evidence from fMNIST is for general optical transformations.)

On p7 you state that the ground truth images are 256 x 256, but fMNIST images are only 28x28. You may have upsampled them to that size, but the information content is much more low-resolution. This is not just a trivial issue, as you will find that the ability to transmit higher-resolution content which is actually using the full resolution in a diverse set of images via this approach will require exponentially increasing amounts of data, unless more physically-informed approaches are taken.

If we can disambiguate the 10 classes with a classifier, we can very easily generate suitable images which approximate the solution. Increasing the latent space we can then generate subgenres of each class which will approximate the solution.

The novel use of multiplexed data as in this experiment certainly complicates the task, as we can have multiple images overlapping and mapping to a single speckle, but this potentially can also help to disambiguate the parameters of the forward mapping from the MMF, so could inspire new work. However, I think the dataset is still too limited to demonstrate the points I think the paper is claiming to make. I predict that if you put in some general natural scenes to the system you will only be able to generate fMNIST-like solutions, so you are not demonstrating generalising multiplexing transmission of optical information in an MMF, as claimed. I believe you are learning to map a significantly reduced subset of classes for a limited domain, from which you can generate an image which approximates the solution. This can have its uses, but I think you need to be much more transparent about it, and the risks involved (e.g. if you use your approach in a medical context, it could generate dangerous false information which looks very convincing to the untrained eye). I think it is a much healthier approach to try to separate the elements of inversion of the MMF pipeline and domain-knowledge-based improvement of the inverse solution, so that end-users have the opportunity to see which aspects are based on raw sensor data, and which aspects make strong assumptions about the nature of the image domain.

In your explanation of the network performance I think you underplay the importance of the FC layer at the start of your network. This is a real-valued (I think) full matrix which approximates the inverse transmission matrix to some degree, and allows for a mapping which could to some degree undo the nonlocal dispersion of the MMF and is vital for a follow-on spatially local convnet architecture to make any sense at all. It is then possible to view your network as a nonlocal, real-valued approximate inversion

matrix, followed by a smoothing, enhancing autoencoder (which can denoise and offer superresolution etc). This sort of approach has already been taken in a paper missing from your paper's bibliography:

J. Mitton, et al., Bessel Equivariant Networks for Inversion of Transmission Effects in Multi-Mode Optical Fibres, NeurIPS 2022.

https://proceedings.neurips.cc/paper_files/paper/2022/file/666dd0d92a64396e753c691db93493d4-Paper-Conference.pdf

https://proceedings.neurips.cc/paper_files/paper/2022/file/666dd0d92a64396e753c691db93493d4-Supplemental-Conference.pdf

this paper extends Camarazza et al's work [ref 24 in your paper], to both bring more prior knowledge into the constraints of the fully connected network, but it also has a network which plays a role similar to the ResNet network in your paper for denoising and post-processing the image towards the ground truth. An important point made in the Mitton et al paper is the danger of the deep neural network filling in information which is not there in the image, and they make the point about the importance of being able to disentangle the attempts to invert the effect of the MMF from the attempts to clean up the image, based on knowledge of the problem domain. If your system makes strong assumptions about the type of data it expects to see, it can't see anything novel to that.

At several points in the paper you use the adjective 'complex'. In each case I believe you intend this to mean 'complicated', but given that the transmission matrix is complex-valued, this usage will possibly confuse people to make them think you are using complex-valued parameters. I recommend changing the term to be clearer.

In general the paper is clearly written and has good quality figures, but would benefit from proof-reading to improve the clarity of the written English.

Reviewer #2 (Remarks to the Author):

In this paper, the authors proposed a deep neural network, termed speckle light field retrieval network (SLRnet), for non-orthogonal optical multiplexing leveraging. Using this SLRnet, the complex mapping relation between two-channel non-orthogonal input light field can be recovered from single-shot speckle output. The experimental results demonstrated that SLRnet can essentially solve the ill-posed problem of non-orthogonal optical multiplexing over a multimode fiber (MMF). Here non-orthogonal optical multiplexing means the multiplexed two-channel wavefronts have the same polarization, wavelength and spatial position. Overall, this paper is generally convincing and straightforward. However, there are several problems need to be addressed before its acceptance.

1. The most fundamental problem is: how does this SLRnet solve the multiple-to-one mapping relationship or the ambiguous solution problem. For example, in this paper, the authors accurately recovered two-channel wavefronts (A)₁ and (B)₂ when the input wavefront is combining (A)₁ and (B)₂. Here ()₁ and ()₂ means two different channels with the same polarization and wavelength. What if the input wavefront is combining (A+B)₁ and (0)₂ or (0.9A+0.5B)₁ and (0.1A+0.5B)₂? Will these three equivalent multiplexing situations produce the same speckle intensity maps? If the speckle intensity maps are the same, can these separate equivalent multiplexing situations be recovered from them? If the output speckle intensity maps are not the same, the authors need to explain in detail why different speckle patterns are output while the same multiplexed wavefront is coupled into the MMF.

2. In Equ. (4), how is T_i defined? In my opinion, there should be only one matrix T to determine the transmission characteristic of the MMF. When we put Equ. (3) into Equ. (1), the resulting output light field should be $E_{out} = T \sum E_i = \sum T E_i \neq \sum T_i E_i$. If the authors defined $T = \sum T_i$, Equ. (4) is still wrong. If there is just one matrix T , maybe this SLRnet could not solve the ambiguous solution problem actually.

3. Are these two input wavefronts fully coherently superposed and multiplexed? The authors only mentioned that these two input wavefronts have the same polarization direction and the same wavelength. If the two channels are fully coherently superimposed, why doesn't this SLRnet directly reconstruct a superimposed monochromatic wavefront?

4. What is the meaning of the grids on the schematic diagram of the input light field at the proximal end of the MMF in Fig. 1(a)?

5. It is recommended that the author modify Fig. 1(a), to emphasize that there are only two multiplexed channels for the input. In this way, it can correspond to only two recovered output channels on the right side of Fig. 1(b).

6. In Figs. 4-5, why did the authors test almost binary images as phase maps? If they don't use the Fashion-MNIST dataset but use phase maps similar to speckle morphology with the gray-scale varies randomly, will the reconstruction results be more erroneous? If the input phase value is larger, exceeding 2π , will the reconstruction results have phase unwrapping problems?

7. Compared with Fig. 4, why are all inputs of the same polarization direction in Fig. 5? Are other orthogonal optical multiplexing cases not tested?

8. Is there a limitation of multiplexing channels? If more than two complex wavefronts with the same polarization direction and the same wavelength are multiplexed, will the resulting error become larger?

Reviewer #3 (Remarks to the Author):

The manuscript entitled “Non-orthogonal optical multiplexing empowered by deep learning” by T. Pan et al reports on an interesting and counterintuitive concept of nonorthogonal optical multiplexing, which is achieved over a 50 m multimode fiber enabled by a deep neural network. The neural network can decode the nonorthogonal spatial overlapping multiplexing channels even though these channels are using the same polarization state and wavelength. The idea of this manuscript is clearly presented and supported by several experimental results. Therefore, I can recommend the publication of this manuscript in Nature Communications if the following comments are addressed by the authors:

Major points:

1. How can the authors achieve phase modulation when the corresponding amplitude is zero? Is there any residual amplitude modulation?
2. Are there any selection rules for the images in the Fashion-MNIST dataset used for training?
3. The pretreatment of the training and validation data set is not extensively discussed. For example, the speckle shown in Figs. 4 and 5 of the main text is a color figure, which is a triple-channel RGB image. However, as can be seen from Fig. 2b, the input of network is a single channel. The scale bar of the speckle images is also required in Fig. 4-5.
4. The demultiplexing results for a 1 m multimode fiber a better than the one of a 50 m multimode fiber. The authors should comment on this issue.

Minor points:

1. The authors are encouraged to provide the open source of the deep neural network, which will substantially increase the impact of this work.
2. Whether the multimode fiber is a step-index multimode fiber or a graded-index multimode fiber? It is not mentioned in the manuscript.
3. There is a ‘?’ in Fig. 1, please check this issue.
4. The full names for ‘Obj’ are inconsistent in the main text and SI.

Reviewer #4 (Remarks to the Author):

The authors of the manuscript entitled 'Non-orthogonal optical multiplexing empowered by deep learning' study the non-orthogonal optical multiplexing over a multimode fiber using a deep neural network. They experimentally demonstrate that even input signals sharing the same polarization, wavelength, and spatial position can be explicitly demultiplexed, where the multichannel amplitude and phase-encoded information can be decoded by using only a single-shot speckle intensity. The concept of non-orthogonal optical multiplexing enabled by deep learning is very interesting, which might provide a new way for transmitting information and find potential applications in short-range optical communication and optical encryption. Therefore, I recommend the publication of this manuscript in Nature Communications with the following comments:

1. As I mentioned above, the multichannel light field information can be decoded by using only a single-shot speckle intensity. What is the role of multiple scattering play in non-orthogonal multiplexing? Whether the multiple scattering processes of the multimode fiber enable this kind of decoding?
2. The fidelity of complex patterns is not as good as the simpler ones. Whether it is limited by the neural network or by the experimental system?
3. The multimode fiber is very sensitive to the perturbation of the environment. How about the robustness of the nonorthogonal optical multiplexing against the perturbation of the environment?
4. The authors use multimode fiber with different core radii and lengths. Which parameters are more important for achieving high-fidelity decoding?
5. Whether the size of the data set will affect the performance of the network?
6. Why they chose the wavelength of 532 nm for the experiment? It is not the standard communication wavelength.

%%%%%%%%%

Response to the Referee 1

%%%%%%%%%

Blue color outlines the original text,

Red color indicates the corresponding modifications

GENERAL COMMENT:

This paper takes a task which has been approached by a number of previous papers (cited in the work), and adds a novel task complication of non-orthogonal multiplexed images and a new combination of deep learning layers which you apply to the standard approximate inversion problem from a speckle image.

The paper poses the task on p1 “If the orthogonal paradigm of optical multiplexing can be break, unprecedented non-orthogonal multiplexing can be achieved, which will become a promising way to meet the challenge of information capacity crunch.” then making the very strong claim “In this work, we show that non-orthogonal optical multiplexing can be explicitly achieved empowered by the SLRnet. As a proof-of-concept experiment, we demonstrate that multiplexing transmission of optical information in a MMF can be realized utilizing non-orthogonal channels”.

Unfortunately, although there is valuable, interesting empirical work in the paper, which could inspire other researchers, I do not believe that the paper justifies such a strong claim, and therefore cannot be published in its current form.

OUR REPLY:

We really thank the Referee for recognizing that out that our work is “a novel task complication of non-orthogonal multiplexed images and a new combination of deep learning layers” and “valuable, interesting empirical work in the paper, which could inspire other researchers”.

We are very sorry for our exaggerated statements in the manuscript, where our previous experimental results cannot support such strong claim as pointed out by the Referee. We delete our exaggerated statements mentioned by the Referee and revise our claim to be more realistic.

In order to address this very important general comment, we change the sentence “If the orthogonal paradigm of optical multiplexing can be break, unprecedented non-orthogonal multiplexing can be achieved, which will become a promising way to meet the challenge of information capacity crunch.” to “If the orthogonal paradigm of optical multiplexing can be break, it could be a step forward for realizing non-orthogonal optical multiplexing, which will become a promising way to meet the challenge of information capacity crunch.”.

We also change the sentence “In this work, we show that non-orthogonal optical multiplexing can be explicitly achieved empowered by the SLRnet. As a proof-of-concept experiment, we demonstrate that multiplexing transmission of optical information in a MMF can be realized utilizing non-orthogonal channels, as schematically shown in Fig. 1c.” to “In this work, we show that preliminary non-orthogonal optical multiplexing through an MMF can be achieved empowered

by the SLRnet. As a proof-of-concept demonstration, multiplexing transmission of information in an MMF, including general natural scene images, uncorrelated QR codes and images not belong to the same type of training dataset, can be realized utilizing non-orthogonal channels, as schematically shown in Fig. 1b, facilitating the realization of non-orthogonal multiplexing transmission of optical information.”.

We also thank the Referee for his/her important comments which help us to substantially improve the quality of our manuscript. According to his/her comments, we perform more experiments to further consolidate the concept proposed in our manuscript. New experimental results for uncorrelated QR codes, general natural scene images, and snapshots in Muybridge recordings not belong to the same type of training dataset are presented. In the following, we address all the important comments one-by-one.

COMMENT 1:

The initial concern builds upon the author’s own observations on p3 “In order to realize demultiplexing of non-orthogonal signals through the MMF from a single-shot output speckle, the inverse mapping H^{-1} of the above equation

$$H^{-1}(I) = Cn \quad (5)$$

should be obtained. There is no physics-based theory reported so far that can effectively obtain H^{-1} when $n > 1$.”

If we cannot effectively find the solution to this clearly ill-posed inverse problem, we need to clearly discuss the assumptions involved in the approximation provided by the neural network. We also need to think carefully about the nature of empirical evidence, and the choice of data to demonstrate ‘multiplexing transmission of optical information in a MMF’ in a convincing manner. What sort of images can this system actually communicate?

OUR REPLY:

We really to thank the Referee for this very important comment. Yes, the Referee is right. As pointed out by the Referee, the dataset we used before is too simple to validate the robustness and generality of the proposed concept. Demonstrating the universal applications of the proposed SLRnet for diverse types of multiplexing images is very important to promote the reported concept of non-orthogonal optical multiplexing over an MMF. In our previous Supplementary Material (Supplementary Note 2), we provided more complex grayscale encoded information from the CelebA face dataset with different human face images. However, they are not as diverse as the natural scene images.

In order to demonstrate the generality of non-orthogonal optical multiplexing through an MMF in a convincing manner, we demonstrate the multiplexed transmission of QR codes that are consisted of uncorrelated digits 0 and 1 first. Such data resembles the most fundamental data in optical communication. The achieved averaged bit accuracy rate in the validation dataset is 98% for four QR codes encoded in two non-orthogonal light field channels. Three typical results are shown in Fig. R1. Such new experimental results indicate that demultiplexing of binary digital information

encoded in non-orthogonal channels is possible using deep learning.

Fig. R1 Demultiplexing results for the non-orthogonal optical multiplexing of uncorrelated binary QR. The speckle outputs, the corresponding ground truths, the retrieved results of neural network, the binarized results and the bit accuracy rate are given, respectively. The length of the MMF is 1m. It should be pointed out that the SLRnet here is revised to facilitate the demultiplexing of the QR codes, as can be seen in our new Supplementary Note 7. The amplitude and phase are self-normalized. The ground truths of uncorrelated QR codes are generated by the uniform random function of Matlab, which consists of 20 x 20 pixels.

More importantly, we follow the Referee’s suggestion to transmit general natural scene images over the MMF. The results will be presented and discussed in the Reply to Comment 2. Our experimental results for the natural scene images show that SLRnet does not rely on the types of transmitted image. The SLRnet can learn the complicated mapping relationship between speckle output and the amplitude and phase distributions of the incident light field.

In order to address this important comment, we change the following sentence in the main text from “The results considering more complex grayscale encoded information from the CelebA face dataset³⁵ further validate that the SLRnet has an excellent ability to retrieve multiplexed information encoded in the non-orthogonal channels (see Supplementary Note 2).” to “In order to showcase the generality of the SLRnet for a diverse set of images, various experimental results considering more complicated grayscale encoded information from the CelebA face dataset³⁵, QR codes whose binary digital information are uncorrelated, general natural scene images from the ImageNet database³⁷, and snapshots in Muybridge recordings not belong to the same type of training dataset are presented (see Supplementary Note 2).”

A new reference is added in the main text.

[37] Deng, J. et al. ImageNet: A large-scale hierarchical image database. In *2009 IEEE Conference on Computer Vision and Pattern Recognition*, 248-255 (2009).

The Fig. R1 is added in the Supplementary Note 2 and the following sentences are added:

“The non-orthogonal multiplexing of uncorrelated QR codes through an MMF is further demonstrated, as shown in Supplementary Figure 3. The averaged bit accuracy rate is about 98%,

validating the potential of applying the reported non-orthogonal optical multiplexing concept for binary data transmission.”

The following sentences and a figure are added in the Supplementary Note 7:

“In order to facilitate the decoding of uncorrelated QR codes, we make minor changes to the SLRnet, where the revised part of network is shown in Supplementary Figure 8. The modified SLRnet uses a ResConvD module instead of the ResConvT module, and eliminates skip connections in the ResUnet. In that case, SLRnet does not perform upsampling after downsampling to a certain size.”

A new figure for the revised SLRnet for the non-orthogonal multiplexing of QR codes is shown in Fig. R2 which is also added in the Supplementary Note 7.

Fig. R2 The modified SLRnet for the non-orthogonal multiplexing of uncorrelated QR codes. **a** It consists of the same base modules, where skipping connections are eliminated. ResConvT is replaced by ResConvD. **b** The configuration of ResConvD.

COMMENT 2:

The dataset chosen fMNIST is a commonly used dataset as an introductory tool in machine learning, but is well-known to be a very simple task, with 10 major classes, and many examples in each class which are very similar to each other. It is also an easy task to create autoencoders which can reproduce the images, and many images in each class are very similar to each other. E.g. a simple UMAP low dimensional embedding can take fMNIST data and generate 10 separate classes in an unsupervised fashion, and can still do this if the images are transformed by random unitary matrices, akin to a MMF transformation). (a fully-connected network would do as well with the raw images as with the unitarily transformed ones. A conv-net would not do as well with the transformed data, given its spatial assumptions, but on fMNIST even a simple fully-connected network gets over 96% accuracy, which highlights how limited evidence from fMNIST is for general optical transformations.)

OUR REPLY:

Yes, the Referee is right. We agree that the fMNIST is too simple to support the proposed concept. Following the Referee’s suggestion, we consider the case of non-orthogonal optical multiplexing through the MMF using general natural scene images from the ImageNet database [Proc. IEEE Conf. Comput. Vis. Pattern Recognit., 248–255 (2009)]. The achieved averaged SSIM/PCC is about 0.737/0.905 in the validation dataset for four natural scene images encoded in two non-orthogonal

multiplexed light field, where the achieved averaged fidelity is comparable to the typical fidelity of state-of-the-art orthogonal multiplexing method using a physically-informed approach [Nat. Commun. 10, 2029 (2019)]. Typical demultiplexing results are shown in Fig. R3. These new experimental results further consolidate that demultiplexing of general natural scene images encoded in non-orthogonal channels is possible using deep learning.

Fig. R3 Demultiplexing results for the non-orthogonal optical multiplexing of general natural scene images. The ground truths, the speckle outputs, and the corresponding retrieved light field information by the SLRnet are shown, respectively, where their corresponding SSIM and PCC are given. Colorbars are also provided for the grayscale images encoded in the amplitude and phase. These images are from the ImageNet database.

In order to address this comment, the following sentences are added in the main text:

“Typical results for general natural scene images are shown in Fig. 6a, where the achieved averaged SSIM/PCC is 0.737/0.905.”

And the following sentences are added in the Supplementary Note 2:

“The non-orthogonal multiplexing of general natural scene images from ImageNet database² is also demonstrated, as shown in Fig. 6a of the main text. The achieved averaged SSIM/PCC is 0.737/0.905, which are comparable to the results of the orthogonal counterpart using physically-informed approach for transmitting natural scene images.”

A new reference is added in the Supplementary Information:

[2] Deng, J. et al. ImageNet: A large-scale hierarchical image database. In *2009 IEEE Conference on Computer Vision and Pattern Recognition*, 248-255 (2009).

COMMENT 3:

On p7 you state that the ground truth images are 256 x 256, but fMNIST images are only 28x28. You may have upscaled them to that size, but the information content is much more low-resolution.

This is not just a trivial issue, as you will find that the ability to transmit higher-resolution content which is actually using the full resolution in a diverse set of images via this approach will require exponentially increasing amounts of data, unless more physically-informed approaches are taken.

If we can disambiguate the 10 classes with a classifier, we can very easily generate suitable images which approximate the solution. Increasing the latent space we can then generate subgenres of each class which will approximate the solution.

OUR REPLY:

We thank the Referee for this very important comment. Yes, the Referee is right. Increasing the resolution of the transmitted information content is not a trivial issue. In our demonstration, the effective resolutions of human face images from the CelebA dataset (see Supplementary Figure 2) and general natural scene images from the ImageNet database (see Fig. R3) are 128 x 128 and 92 x 92, respectively, indicating that the proposed non-orthogonal optical multiplexing can also be achieved for higher-resolution images. Furthermore, because we realize complex modulation (amplitude and phase) using a phase-only SLM, the modulation precision is limited compared with the pure phase modulation considering the residual phase modulation error of a commercial SLM [New J. Phys. 13, 123021 (2011)]. If we achieve the non-orthogonal optical multiplexing only by the phase encoding, better averaged retrieved fidelity (SSIM/PCC, 0.819/0.945) under the same resolution can be achieved, where three typical results are shown in Fig. R4.

Fig. R4 Demultiplexing results for the non-orthogonal optical multiplexing of general natural scene images using the phase dimension only. The ground truths, the speckle outputs, and the corresponding retrieved phases of light field information by the SLRnet are shown, respectively, where their corresponding SSIM and PCC are given. Colorbars are also provided. These images in **a** and **b** are from the ImageNet database and snapshots from Muybridge recordings from the 1870s that marked the historically important breakthrough of the first ever high-speed photography images. The data in **(b)** is not belong to the training dataset of the SLRnet, i. e. ImageNet database.

As pointed out by the Referee, changing the transmitted information content to a more diverse set of images will need more amounts of data. In our demonstration, we can achieve comparable SSIM (0.819) for non-orthogonal optical multiplexing using the same amounts of data (50000 images) as the physically-informed approach without non-orthogonal multiplexing [Nat. Commun. 10, 2029 (2019)] by using the SLRnet. As suggested by the Referee, the demultiplexing fidelity can be further improved and the required data scale can be reduced by combing a physically-informed model of the multiple scattering system with the neural network. We will address this very inspired suggestion in our future work.

We would like to thank the Referee again for this inspired comment, which drive us to try the non-orthogonal multiplexing of general natural scene images. We also test images not belong to the training dataset to show the generalization of the proposed neural network. It is anticipated that these new experimental results of general natural scene images, uncorrelated QR codes and images from a video could further support the concept proposed in our manuscript.

In order to address this comment, Figs. R3 and R4 are added in the main text as a new Fig. 6.

We also add the following sentences in the main text:

“To further increase the modulation precision of wavefront, the information is only encoded in the phase dimension of two non-orthogonal channels. The achieved averaged fidelity can be up to 0.819/0.945 (SSIM/PCC) as shown in Fig. 6b, which is substantially improved compared with the complex modulation case. At the same time, the achieved typical fidelity for images not belong to the ImageNet database can be up to 0.88/0.97 (SSIM/PCC), as shown in Fig. 6c, indicating good generalization of the SLRnet. In addition, adding a physically-informed model of the MMF system in the deep neural network could be an effective mean for further boosting the demultiplexing fidelity^{24,25}.”

At the same time, the following sentences are added in the Supplementary Note 2:

“Because the amplitude and phase modulations are applied using a phase-only SLM, the modulation precision is limited. If we only encode the information in the phase dimension, the achieved averaged fidelity can be up to 0.819/0.945 (SSIM/PCC), which is substantially improved compared with the complex modulation case. Typical results for the non-orthogonal multiplexing of general natural scene images from ImageNet database² and snapshots from Muybridge recordings are shown in Fig. 6b and c of the main text, respectively. It should be emphasized that the data from the Muybridge recordings (see Fig. 6c) is not belong to the same type of training dataset of the neural network.”

COMMENT 4:

The novel use of multiplexed data as in this experiment certainly complicates the task, as we can have multiple images overlapping and mapping to a single speckle, but this potentially can also help to disambiguate the parameters of the forward mapping from the MMF, so could inspire new work. However, I think the dataset is still too limited to demonstrate the points I think the paper is claiming to make. I predict that if you put in some general natural scenes to the system you will only be able to generate fMNIST-like solutions, so you are not demonstrating generalising multiplexing transmission of optical information in an MMF, as claimed. I believe you are learning to map a significantly reduced subset of classes for a limited domain, from which you can generate an image which approximates the solution. This can have its uses, but I think you need to be much more transparent about it, and the risks involved (e.g. if you use your approach in a medical context, it could generate dangerous false information which looks very convincing to the untrained eye). I think it is a much healthier approach to try to separate the elements of inversion of the MMF pipeline and domain-knowledge-based improvement of the inverse solution, so that end-users have the opportunity to see which aspects are based on raw sensor data, and which aspects make strong

assumptions about the nature of the image domain.

OUR REPLY:

Yes, the Referee is right. We also thank the Referee for recognizing that our results “could inspire new work”.

Following the suggestion of the Referee, we have achieved comparable retrieved fidelity in non-orthogonal optical multiplexing of general natural scene images and snapshots from Muybridge recordings as previous state-of-the-art approaches without non-orthogonal multiplexing, as shown in Figs. R3 and R4. We also demonstrate that the non-orthogonal optical multiplexing of uncorrelated QR codes, whose averaged bit accuracy rate can be up to 98%, as shown in Fig. R1. Our experimental results for the natural scene images, QR codes and snapshots of Muybridge recordings show that the proposed neural network is quite robust to different types of images. The neural network can learn the complicated mapping relationship between a single-shot speckle output and the amplitude and phase distributions of two incident light field channels. It is anticipated that all the results presented in the revised manuscript could be a step forward for facilitating non-orthogonal multiplexing transmission of information through an MMF.

We fully agree that we need to be much more transparent about the approach presented in this manuscript and clearly outline the risks involved to the audience. We also agree that our approach cannot be used in a medical context at this stage because the achieved averaged fidelity of demultiplexing general natural scene images cannot approach unity.

In order to address this comment, we change the following sentence in the main text from “...facilitating the unprecedented multiplexing of non-orthogonal channels for massive transmission of optical information.” to “**Although the proposed concept of non-orthogonal optical multiplexing over an MMF cannot be directly used in medical diagnosis at this stage, which requires unity fidelity, the non-orthogonal multiplexing of uncorrelated binary digital information with high accuracy indicates a step forward for realizing non-orthogonal multiplexing transmission of optical information through an MMF.**”

The following sentence “**In addition, adding a physically-informed model of the MMF system in the deep neural network could be an effective mean for further boosting the demultiplexing fidelity^{24,25}.**” is added in the main text.

COMMENT 5:

In your explanation of the network performance I think you underplay the importance of the FC layer at the start of your network. This is a real-valued (I think) full matrix which approximates the inverse transmission matrix to some degree, and allows for a mapping which could to some degree undo the nonlocal dispersion of the MMF and is vital for a follow-on spatially local convnet architecture to make any sense at all. It is then possible to view your network as a nonlocal, real-valued approximate inversion matrix, followed by a smoothing, enhancing autoencoder (which can denoise and offer superresolution etc). This sort of approach has already been taken in a paper

missing from your paper's bibliography:

J. Mitton, et al., Bessel Equivariant Networks for Inversion of Transmission Effects in Multi-Mode Optical Fibres, NeurIPS 2022.

https://proceedings.neurips.cc/paper_files/paper/2022/file/666dd0d92a64396e753c691db93493d4-Paper-Conference.pdf

https://proceedings.neurips.cc/paper_files/paper/2022/file/666dd0d92a64396e753c691db93493d4-Supplemental-Conference.pdf

this paper extends Camarazza et al's work [ref 24 in your paper], to both bring more prior knowledge into the constraints of the fully connected network, but it also has a network which plays a role similar to the ResNet network in your paper for denoising and post-processing the image towards the ground truth. An important point made in the Mitton et al paper is the danger of the deep neural network filling in information which is not there in the image, and they make the point about the importance of being able to disentangle the attempts to invert the effect of the MMF from the attempts to clean up the image, based on knowledge of the problem domain. If your system makes strong assumptions about the type of data it expects to see, it can't see anything novel to that.

OUR REPLY:

Yes, the Referee is right. We really thank the Referee for drawing our attention to this very related and important reference we missed in our previous manuscript. We fully agree with the Referee's elegant explanation of the role for the FC layer played in our neural network. The introduction of the FC layer effectively enhances the global connection of the network and compensates the lack of non-local ability of convolutional layer due to local shared receptive field. At the same time, the number of network's output channels can be flexibly manipulated by the convolutional layer.

In our previous manuscript, the transmitted data type is too trivial that cannot fully support our argument. Following the suggestions of the Referee, we demonstrate non-orthogonal optical multiplexing of general natural scene images and uncorrelated binary QR codes in Figs. R1, R3 and R4 (a). More importantly, we also test the demultiplexing results for the images not belong to the same type of training dataset of the neural network, as shown in Fig. R4 (b). These images are snapshots from Muybridge recordings from the 1870s. As can be seen from these figures, the retrieved typical fidelity of SSIM/PCC is up to 0.88/0.97, indicating good generalization of the neural network. Therefore, the proposed SLRnet can be applicable to a diverse set of images encoding in non-orthogonal multiplexing input channels.

In order to address this comment, we add the following reference and make suitable citation:

[25] Mitton, J. et al. Bessel equivariant networks for inversion of transmission effects in multi-mode optical fibres. *Adv. Neural Inf. Process. Syst.* 35, 16010–16022 (2022).

We revise the following sentences from "It consists of a fully connected (FC) layer and ResUnet³⁴, whose main advantages over Uet are as follows: (1) a FC layer is introduced before the input of Unet to enhance the fitting and generalization ability of the network. The introduction of the FC layer improves the performance of demultiplexing multidimensional encoded information using a single speckle output; (2) a large number of skip connections are introduced in the encoder-decoder path to enhance the degeneration-free propagation of data in the network (See "Methods" section

for details).” to “It consists of a fully connected (FC) layer and ResUnet³⁴, whose main advantages over Unet are as follows: (1) a FC layer is introduced before the input of Unet to enhance the fitting and generalization ability of the network. The introduction of the FC layer can effectively undo the nonlocal dispersion of the MMF, which improves the performance of demultiplexing multidimensional encoded information using a single speckle output. The ResUnet is used for denoising and post-processing the multiplexing information towards the ground truth, which is similar to the convnet proposed recently²⁵. In addition, the convolutional layer can also facilitate the manipulation of multichannel outputs in the non-orthogonal multiplexing without increasing of training burden; (2) a large number of skip connections are introduced in the encoder-decoder path to enhance the degeneration-free propagation of data in the network (See "Methods" section for details).”

We also add the following sentence “In addition, adding a physically-informed model of the MMF system in the deep neural network could be an effective mean for further boosting the demultiplexing fidelity^{24,25}.”

The sentence “Specifically, deep neural network has been used to improve the performance of orthogonal multiplexing over a multiple scattering medium²²⁻³². To date, however, all the reported multiplexing scenarios strictly relies on the physical orthogonality among multiplexing channels^{9-13, 22-32}.” is changed to “Specifically, deep neural network has been used to improve the performance of orthogonal multiplexing over a multiple scattering medium²²⁻³³. To date, however, all the reported multiplexing scenarios strictly relies on the physical orthogonality among multiplexing channels^{11-15, 22-33}.”

COMMENT 6:

At several points in the paper you use the adjective ‘complex’. In each case I believe you intend this to mean ‘complicated’, but given that the transmission matrix is complex-valued, this usage will possibly confuse people to make them think you are using complex-valued parameters. I recommend changing the term to be clearer.

OUR REPLY:

We thank the Referee for pointing out this unclear presentation in our manuscript. In order to address this comment, we revise them accordingly which are outlined in the red-marked PDF.

COMMENT 7:

In general the paper is clearly written and has good quality figures, but would benefit from proof-reading to improve the clarity of the written English.

OUR REPLY:

In order to address this comment, we carefully checked the whole manuscript to improve the English presentation. The revised parts are outlined by red in the marked PDF file.

In summary, we really thank the Referee for these important comments which substantially help us

to improve the quality of our manuscript. We are very sorry for the exaggerated statements in our previous manuscript, where such claim is revised to be more realistic. The new experimental results using the general natural scene images, uncorrelated QR codes, and images not belong to the same type of training dataset are presented to consolidate our claim. It is anticipated that the revised manuscript is now can meet the criteria of Nature Communication and your consideration is sincerely appreciated.

%%%%%%%%%

Response to the Referee 2

%%%%%%%%%

Blue color outlines the original text

Red color indicates the corresponding modifications

GENERAL COMMENT:

In this paper, the authors proposed a deep neural network, termed speckle light field retrieval network (SLRnet), for non-orthogonal optical multiplexing leveraging. Using this SLRnet, the complex mapping relation between two-channel non-orthogonal input light field can be recovered from single-shot speckle output. The experimental results demonstrated that SLRnet can essentially solve the ill-posed problem of non-orthogonal optical multiplexing over a multimode fiber (MMF). Here non-orthogonal optical multiplexing means the multiplexed two-channel wavefronts have the same polarization, wavelength and spatial position. Overall, this paper is generally convincing and straightforward. However, there are several problems need to be addressed before its acceptance.

OUR REPLY:

We really thank the Referee for pointing out our manuscript is “Overall, this paper is generally convincing and straightforward”. We are sorry for our confused presentation in our previous manuscript. Non-orthogonal optical multiplexing in our manuscript means that the multiplexed channels have non-orthogonal polarizations, the same wavelength, and the same spatial position. Two multiplexing channels with the same polarization, wavelength and spatial position is the extreme case of non-orthogonal optical multiplexing.

In order to address this comment, we add the following sentence in the main text:

“While non-orthogonal optical multiplexing over an MMF can be referred to multiplexing input channels possessing non-orthogonal polarizations, the same wavelength, and the same spatial position, where their polarizations are even the same for the typical non-orthogonal scenario.”

We also address all his/her important comments point by point in the following.

COMMENT 1:

The most fundamental problem is: how does this SLRnet solve the multiple-to-one mapping relationship or the ambiguous solution problem. For example, in this paper, the authors accurately recovered two-channel wavefronts (A)1 and (B)2 when the input wavefront is combining (A)1 and (B)2. Here ()1 and ()2 means two different channels with the same polarization and wavelength. What if the input wavefront is combining (A+B)1 and (0)2 or (0.9A+0.5B)1 and (0.1A+0.5B)2? Will these three equivalent multiplexing situations produce the same speckle intensity maps? If the speckle intensity maps are the same, can these separate equivalent multiplexing situations be recovered from them? If the output speckle intensity maps are not the same, the authors need to explain in detail why different speckle patterns are output while the same multiplexed wavefront is coupled into the MMF.

OUR REPLY:

We would like to thank the Referee for this very important comment. The output speckle intensity maps are not the same for the scenarios pointed out by the Referee even for channels using the same polarization. This is because there is residual asymmetry in the optical paths of two multiplexing input channels before the pupil plane of the first objective lens. The residual asymmetry mainly comes from imperfect parallelism of two multiplexed beams and imperfect parallelism of two polarizations, etc. Slightly different input wavefronts before the pupil plane of Obj₂ will be obtained for the input wavefronts of (A)₁&(B)₂ and (A+B)₁&(0)₂. As a result, the excited modes of the MMF will not be exactly the same. This difference will be further leveraged by the multiple scattering of the MMF, resulting in distinct output speckle patterns. In order to support our argument, we show the corresponding output speckle patterns of our experimental setup for the typical cases of (A)₁&(B)₂ and (A+B)₁&(0)₂ pointed out by the Referee as an example. Without loss of generality, we consider the case of amplitude modulation only for the convenience of demonstrating (A+B). As can be seen from Fig. R5, the corresponding output speckle patterns are different for two input scenarios of (A)₁&(B)₂ and (A+B)₁&(0)₂. We further using the SLRnet to retrieve these two input wavefronts using their corresponding speckle patterns, where the results are also shown in Fig. R5. As can be seen from these figures, the SLRnet can effectively decode two input wavefronts. It means that the residual optical asymmetry and multiple scattering of the MMF play important roles in solving the multiple-to-one mapping relationship in the MMF. The other cases pointed out by the Referee are similar and not shown here.

Fig. R5 The ground truths, corresponding output speckles of the MMF and the retrieved wavefronts by the SLRnet for the input scenarios of amplitude modulation (A)₁&(B)₂ and (A+B)₁&(0)₂. The SSIM/PCC between two speckle patterns is also given.

In order to address this very important comment, we add the following sentences in the main text: “It should be pointed out that the transmission matrix T_i is different even for the multiplexing channels with parallel polarizations because of the residual optical asymmetry during multiplexing

and coupling to the MMF, such as slightly different k-vectors of the multiplexed beams.” “At the same time, the residual asymmetry of the optical paths plays an important role in the non-orthogonal optical multiplexing through the MMF. The residual asymmetry will be leveraged by the multiple scattering of MMF, resulting in distinct speckle outputs for slightly different inputs, which can facilitate the solution of multiple-to-one mapping relationship.”

COMMENT 2:

In Equ. (4), how is T_i defined? In my opinion, there should be only one matrix T to determine the transmission characteristic of the MMF. When we put Equ. (3) into Equ. (1), the resulting output light field should be $E_{out} = T \sum E_i = \sum T E_i \neq \sum T_i E_i$. If the authors defined $T = \sum T_i$, Equ. (4) is still wrong. If there is just one matrix T , maybe this SLRnet could not solve the ambiguous solution problem actually.

REPLY:

We would like to thank the Referee for this very important comment. We are very sorry for the misunderstanding caused by our unclear presentation. T_i indicates the transmission matrix for a given multiplexing channel mediated by the MMF, which depends on the polarization of the incident light, the k-vector of the collimated incident beam and the incident position at the entrance pupil. For example, the transmission matrix is polarization-dependent for each multiplexing channel. Specifically, even the transmission matrixes of the MMF for two multiplexing light field channels with the same polarization are slightly different in experiment though we fine tune the parallelism of two beams, which can be attributed to the parasitic optical asymmetry in two optical paths of multiplexing input channels. There are also other kinds of residual asymmetry, such as parallelism of two polarizations and imperfect spatial overlap of two beams at the entrance pupil of the first objective lens. According to our new experimental results, this residual asymmetry indeed facilitates the non-orthogonal optical multiplexing of diverse types of images.

Fig. R6 The difference of two measured transmission matrixes for the two non-orthogonal optical multiplexing channels with the same targeted polarization in our experimental setup, where their amplitude and phase are indicated by intensity and colored code, respectively.

This argument can be further verified by experimentally measuring the transmission matrixes for two channels with the same targeted input polarization using a well-established four-step phase shift

method [Phys. Rev. Lett. 104, 100601 (2010)], respectively. The difference of two measured transmission matrixes is shown in Fig. R6, which indicates the transmission matrixes for these two channels are different.

In order to address this comment, we add the following sentence in the main text “It should be pointed out that the transmission matrix T_i is different even for the multiplexing channels with parallel polarizations because of the residual optical asymmetry during multiplexing and coupling to the MMF, such as slightly different k-vectors of the multiplexed beams.”

The sentences in the main text “The input-output relationship of a MMF can be described by a transmission matrix, as shown by the following equation:” is changed to “The single channel input-output relationship of a MMF can be described by a transmission matrix, as shown by the following equation:”.

“Here, T is the transmission matrix ...” is changed to “Here, T is the transmission matrix for a multiplexing channel through the MMF with a given input polarization ...”.

COMMENT 3:

Are these two input wavefronts fully coherently superposed and multiplexed? The authors only mentioned that these two input wavefronts have the same polarization direction and the same wavelength. If the two channels are fully coherently superimposed, why doesn't this SLRnet directly reconstruct a superimposed monochromatic wavefront?

OUR REPLY:

We would like to thank the Referee for this very important comment which helps us to further clarify the process of demultiplexing.

Yes, two input wavefronts are fully coherently superimposed. At the same time, two input wavefronts can also possess distinct polarization states, as shown in Fig. 2 (a) and Fig. 3 of our manuscript. Because we use the complex modulation (amplitude and phase) in our manuscript for encoding multiple information in two input wavefronts simultaneously, the electric fields of two inputs are $\text{Real}(A(x,y)e^{i(\omega t - \mathbf{k} \cdot \mathbf{r} - \varphi A(x,y))})$ and $\text{Real}(B(x,y)e^{i(\omega t - \mathbf{k} \cdot \mathbf{r} - \varphi B(x,y))})$ considering the ideal case of perfect parallel k vectors for simplicity. In this case, the multiplexed information is encoded in the spatial distribution of $A(x,y)$, $\varphi A(x,y)$, $B(x,y)$, and $\varphi B(x,y)$ respectively. The electric field of the superimposed monochromatic wavefront becomes $E(x,y)\cos(\omega t - \mathbf{k} \cdot \mathbf{r} + \varphi(x,y))$,

$$\text{where } E(x,y) = \sqrt{A(x,y)^2 + B(x,y)^2 + 2A(x,y)B(x,y)\cos(\varphi A(x,y) - \varphi B(x,y))}$$

$$\text{and } \varphi(x,y) = \arctan\left(\frac{A(x,y)\sin(\varphi A(x,y)) + B(x,y)\sin(\varphi B(x,y))}{A(x,y)\cos(\varphi A(x,y)) + B(x,y)\cos(\varphi B(x,y))}\right)$$

From the multiplexing and demultiplexing points of view, the information encoded in the amplitude and phase dimensions [i. e. $A(x,y)$, $\varphi A(x,y)$, $B(x,y)$, and $\varphi B(x,y)$] will mix with each other if we directly recover the amplitude $E(x,y)$ and phase distributions $\varphi(x,y)$ of the superimposed wavefront. In order to address this comment, we change the sentences in the main text “They are superimposed with a non-polarized beam splitter cube (NPBS), ...” to “They are coherently superimposed with a

non-polarized beam splitter cube (NPBS), ...”.

“Each light field channel is composed of spatially encoded information in both amplitude $A(x,y)$ and phase $\varphi(x,y)$, respectively, resulting totally four multiplexing channels.” is changed to “Each light field channel is composed of spatially encoded information in both amplitude $A(x,y)$ and phase $\varphi(x,y)$ dimensions, respectively, resulting totally four multiplexing channels for transmitting independent information.”

COMMENT 4:

What is the meaning of the grids on the schematic diagram of the input light field at the proximal end of the MMF in Fig. 1(a)?

OUR REPLY:

We thank the Referee for this comment. The grids of our Fig. 2 (a) schematically indicate the information unit during encoding information in both amplitude $A(x,y)$ and phase $\varphi(x,y)$ dimensions. They are corresponding to the pixels of the transmitted image.

In order to address this comment, we add a sentence “The grids superimposed on the input information indicate the information units in both amplitude and phase dimensions.” in the caption of Fig. 2.

COMMENT 5:

It is recommended that the author modify Fig. 1(a), to emphasize that there are only two multiplexed channels for the input. In this way, it can correspond to only two recovered output channels on the right side of Fig. 1(b).

OUR REPLY:

Fig. R7 The revised Fig. 2 of our manuscript.

Fig. R8 The revised Fig. 1 of our manuscript.

We thank the Referee for this important suggestion. According to Comment 4, we guess the Referee means the content in Fig. 2 of our manuscript. We revise the figure according to the Referee’s suggestion, as shown in Fig. R7. At the same time, we also revise the channel numbers shown in Fig. 1 of our manuscript to avoid confusion. At the same time, we delete previous Fig. 1 (a) to present our motivation straightforwardly and clearly, as shown in Fig. R8.

At the same time, the following sentences in Introduction are changed accordingly. “Considering the demultiplexing of multiple orthogonal signals, physical based theory, such as transmission matrix^{9–13}, can tackle this issue even over a strongly scattering medium. As schematically shown in Fig. 1a, the transmission matrix based methods can decode the orthogonal multiplexing signals mediated by orthogonal polarizations and distinct wavelengths, where full-field measurement of output signal is required. It means that the division nature and the orthogonal paradigm of these multiplexing mechanisms inevitably impose an upper limit of multiplexing capacity^{1–8, 14, 15}. In contrast, these physical based methods fail to decode the non-orthogonal multiplexing signals using the same polarization and wavelength, as schematically shown in Fig. 1b.” is changed to “**However**, the division nature and the orthogonal paradigm of these multiplexing mechanisms inevitably impose an upper limit of multiplexing capacity^{1–10}. Considering the demultiplexing of multiple orthogonal signals, **the transmission matrix method^{11–15}** can tackle this issue even over a strongly scattering medium, **such as an MMF. While non-orthogonal optical multiplexing over an MMF can be referred to multiplexing input channels possessing non-orthogonal polarizations, the same wavelength, and the same spatial position, where their polarizations are even the same for the typical non-orthogonal scenario. In this case, the inverse transmission matrix method fails to decode the multiplexing signals with the same polarization and wavelength using a single-shot intensity detection**, as schematically shown in Fig. 1a.”

COMMENT 6:

In Figs. 4-5, why did the authors test almost binary images as phase maps? If they don't use the Fashion-MNIST dataset but use phase maps similar to speckle morphology with the gray-scale varies randomly, will the reconstruction results be more erroneous? If the input phase value is larger,

exceeding 2π , will the reconstruction results have phase unwrapping problems?

OUR REPLY:

We really thank the Referee for this important comment. We did try more complex images with large gray-scale variation. As shown in the Supplementary Figure 2 of our manuscript, the retrieved fidelity of the human images from the CelebA dataset [In Computer Vision–ECCV 2020: 16th European Conference, Glasgow, UK, August 23–28, 2020, Proceedings, Part XII 16, 70–85] are a little bit lower than the case of the Fashion-MNIST dataset. We also try more general natural scene images from the ImageNet database [Proc. IEEE Conf. Comput. Vis. Pattern Recognit., 248–255 (2009)] in the revised manuscript, as shown in Fig. R9. The averaged achieved SSIM/PCC is 0.73/0.90. The fidelity can be improved by either increasing the amounts of data or using a physically-informed approach. At the same time, because the modulation precision in the wavefront is limited by the complex modulation scheme (amplitude and phase) using a phase-only SLM. If we achieve the non-orthogonal optical multiplexing only by the phase encoding, better average retrieved fidelity (SSIM, 0.819) can be achieved, as shown in Fig. R10 (a). The typical retrieved fidelity can be up to 0.88 (SSIM) even for images not belong to the same type of training dataset, where typical results are shown in Fig. R10 (b). These results further validate the robustness and generalization of the proposed neural network for the non-orthogonal multiplexing of diverse images.

Fig. R9 Demultiplexing results for the non-orthogonal optical multiplexing of general natural scene images. The ground truths, the speckle outputs, and the corresponding retrieved light field information by the SLRnet are shown, respectively, where their corresponding SSIM and PCC are given. Colorbars are also provided for the grayscale images encoded in the amplitude and phase. These images are from the ImageNet database.

Fig. R10 Demultiplexing results for the non-orthogonal optical multiplexing of general natural scene images using the phase dimension only. The ground truths, the speckle outputs, and the corresponding retrieved phases of light field information by the SLRnet are shown, respectively, where their corresponding SSIM and PCC are given. Colorbars are also provided. These images in **a** and **b** are from the ImageNet database and snapshots from Muybridge recordings from the 1870s that marked the historically important breakthrough of the first ever high-speed photography images. The data in **(b)** is not belong to the training dataset of the SLRnet, i. e. ImageNet database.

Because we define the range of modulated phase within $0-2\pi$. Therefore, there is no phase unwrapping problem in our case.

In order to address this comment, Figs. R9 and R10 are added in the main text as a **new Fig. 6**.

We also add the following sentences in the main text:

“To further increase the modulation precision of wavefront, the information is only encoded in the phase dimension of two non-orthogonal channels. The achieved averaged fidelity can be up to 0.819/0.945 (SSIM/PCC) as shown in Fig. 6b, which is substantially improved compared with the complex modulation case. At the same time, the achieved typical fidelity for images not belong to the ImageNet database can be up to 0.88/0.97 (SSIM/PCC), as shown in Fig. 6c, indicating good generalization of the SLRnet. In addition, adding a physically-informed model of the MMF system in the deep neural network could be an effective mean for further boosting the demultiplexing fidelity^{24,25}.”

At the same time, the following sentences are added in the Supplementary Note 2.

“The non-orthogonal multiplexing of general natural scene images from ImageNet database² is also demonstrated, as shown in Fig. 6a of the main text. The achieved averaged SSIM/PCC is 0.737/0.905, which are comparable to the results of the orthogonal counterpart using physically-informed approach for transmitting natural scene images.”

“Because the amplitude and phase modulations are applied using a phase-only SLM, the modulation precision is limited. If we only encode the information in the phase dimension, the achieved averaged fidelity can be up to 0.819/0.945 (SSIM/PCC), which is substantially improved compared with the complex modulation case. Typical results for the non-orthogonal multiplexing of general natural scene images from ImageNet database² and snapshots from Muybridge recordings are shown in Fig. 6b and c of the main text, respectively. It should be emphasized that the data from the Muybridge recordings (see Fig. 6c) is not belong to the same type of training dataset of the neural

network.”

COMMENT 7:

Compared with Fig. 4, why are all inputs of the same polarization direction in Fig. 5? Are other orthogonal optical multiplexing cases not tested?

OUR REPLY:

We would like to thank the Referee for this comment. This is because we want to emphasize the extreme case of non-orthogonal optical multiplexing, i. e. multiplexing channels possessing the same polarization state. Other optical multiplexing cases are similar to the cases of Fig. 3.

In order to address this comment, the sentence “To further consolidate the superiority of SLRnet in a more realistic scenario, the non-orthogonal optical multiplexing results over a 50 m MMF are presented (See "Methods" section for details), as shown in Fig. 5.” is changed to “To further consolidate the superiority of SLRnet in a more realistic scenario, the non-orthogonal optical multiplexing results using the same polarization state over a 50 m MMF are presented (See "Methods" section for details), as shown in Fig. 5. Other cases using different polarization states are similar and not shown here.”

COMMENT 8:

Is there a limitation of multiplexing channels? If more than two complex wavefronts with the same polarization direction and the same wavelength are multiplexed, will the resulting error become larger?

OUR REPLY:

We would like to thank the Referee for this very inspired comment. We believe that the number of multiplexing channels is limited by the modes of MMF as well as the active spatial modulated range of the SLM. The number of modes supported by a multimode fiber always has an upper limit regarding to its core diameter and contrast of refractive indexes. If there are enough modes of the MMF for multiplexing the information, then the multiplexing channels we can demonstrate is limited by the active area of the SLM. At this stage, we divide the SLM into two sub-ranges and the active area of our commercial SLM is limited (1920 x 1080 pixels), which hampers us from multiplexed more channels in experiment.

If more than two complex wavefronts with the same polarization direction and the same wavelength are multiplexed, the resulting error might become larger if we do not enlarge the amount of data simultaneously. Using a physically-informed approach might solve this problem, which we will try to address in our future work.

In order to address this comment, we add the following sentence in the main text “If more than two channels are involved in the non-orthogonal optical multiplexing, the total amount of data should be increased for achieving similar fidelity.”

In summary, we really thank the Referee for these important comments which substantially help us to improve the quality of our manuscript. The clarification of multiple-to-one mapping relationship is very important for our manuscript. It is anticipated that the revised manuscript is now can meet the criteria of Nature Communication and your consideration is sincerely appreciated.

%%%%%%%%%

Response to the Referee 3

%%%%%%%%%

Blue color outlines the original text

Red color indicates the corresponding modifications

GENERAL COMMENT:

The manuscript entitled “Non-orthogonal optical multiplexing empowered by deep learning” by T. Pan et al reports on an interesting and counterintuitive concept of nonorthogonal optical multiplexing, which is achieved over a 50 m multimode fiber enabled by a deep neural network. The neural network can decode the nonorthogonal spatial overlapping multiplexing channels even though these channels are using the same polarization state and wavelength. The idea of this manuscript is clearly presented and supported by several experimental results. Therefore, I can recommend the publication of this manuscript in Nature Communications if the following comments are addressed by the authors:

OUR REPLY:

We would like to thank the Referee for his/her positive comments “The idea of this manuscript is clearly presented and supported by several experimental results” and support of publication in Nature Communications. The comments are very helpful for us to substantially improve the quality of our manuscript. We are sorry for our unclear presentation in our previous manuscript. We address his/her important comments point by point in the following.

COMMENT 1:

How can the authors achieve phase modulation when the corresponding amplitude is zero? Is there any residual amplitude modulation?

OUR REPLY:

We would like to thank the Referee for this important comment. We did make a mistake in labeling the color bars of the figures. The normalized amplitude modulation range is from 0.2-1, which means that all the images encoded in the amplitude dimension are scaled from 0-1 to 0.2-1. The smallest amplitude is not zero for achieving the corresponding phase modulation.

In order to address this important comment, we revise all the color bars for the amplitude without changing the data of these figures.

We also add the following sentence “**All the images encoded in the amplitude dimension are scaled from 0-1 to 0.2-1.**”

COMMENT 2:

Are there any selection rules for the images in the Fashion-MNIST dataset used for training?

OUR REPLY:

We thank the Referee for this important comment. We use 90% of the data in the Fashion-MNIST dataset for training and the other 10% of the data for validating. The data of validation set is uniformly sampled in the Fashion-MNIST dataset, where the remain data forms the training dataset. The selection rules for other datasets are similar.

In order to address this important comment, we change the following sentence “These data are divided into a training set (90%) and a validation set (10%) . The data in validation set is uniformly sampled in the learning data set.” to “Each dataset is divided into a training set (90%) and a validation set (10%) . The data in the validation set is uniformly sampled in its corresponding dataset.”

COMMENT 3:

The pretreatment of the training and validation data set is not extensively discussed. For example, the speckle shown in Figs. 4 and 5 of the main text is a color figure, which is a triple-channel RGB image. However, as can be seen from Fig. 2b, the input of network is a single channel. The scale bar of the speckle images is also required in Fig. 4-5.

OUR REPLY:

We would also like to thank the Referee for this important comment. We find that it is our unclear presentation that leads to this confusion. The speckle patterns shown in Figs. 4 and 5 of the main text are effectively single channel because a CW laser is used as a source for encoding information in non-orthogonal multiplexing channels. The speckle patterns presented in the manuscript are captured by a color CMOS camera. They will be translated to a grayscale image which will then be fed to the neural network.

In order to avoid confusion, we add the following sentence “The captured speckles are translated to a grayscale image.”.

The scale bars are added in Fig. 4-5.

COMMENT 4:

The demultiplexing results for a 1 m multimode fiber is better than the one of a 50 m multimode fiber. The authors should comment on this issue.

OUR REPLY:

We would also like to thank the Referee for this very important comment. During the preparation of datasets, the scattering properties of the multimode fiber are slightly perturbed by the environment. A longer MMF is much more sensitive to various vibrations from the experimental environment than a shorter MMF. Therefore, the demultiplexing results using the neural network for the 1 m multimode fiber is better than the case of 50 m.

In order to address this important comment, we add the following sentences in the main text “As can be seen in Figs. 4 and 5, the demultiplexing results of a 1 m MMF is better than the 50 m case.

This is because the scattering properties of a longer MMF is much easier to be affected by the environment. The demultiplexing performance can be further improved by optimizing the network architecture.”

“In addition, adding a physically-informed model of the MMF system in the deep neural network could be an effective mean for further boosting the demultiplexing fidelity^{24,25}.”

COMMENT 5:

The authors are encouraged to provide the open source of the deep neural network, which will substantially increase the impact of this work.

OUR REPLY:

We thank the Referee for this important suggestion. We provide the open source of the neural network in the following link: <https://github.com/withinWolfcontrol/Non-orthogonal-optical-multiplexing-empowered-by-deep-learning>.

COMMENT 6:

Whether the multimode fiber is a step-index multimode fiber or a graded-index multimode fiber? It is not mentioned in the manuscript.

OUR REPLY:

Both multimode fibers we used are step-index multimode fibers. In order to address this comment, we add the following sentence “Both MMFs are step-index MMFs.” in Methods.

COMMENT 7:

There is a ‘?’ in Fig. 1, please check this issue.

OUR REPLY:

Fig. R11 The revised Fig. 1 of our manuscript.

We are sorry for this misprint. We delete this symbol and revise Fig. 1 based on the comment of another Referee as well, as shown in Fig. R11.

COMMENT 8:

The full names for ‘Obj’ are inconsistent in the main text and SI.

OUR REPLY:

In order to address this comment, “**Microscopic objective**” in supplementary material is changed to “**objective lens**”

In summary, we really thank the Referee for these important comments which substantially help us to substantially shape our manuscript. It is anticipated that the revised manuscript now can meet the criteria of Nature Communication and your consideration is sincerely appreciated.

%%%%%%%%%

Response to the Referee 4

%%%%%%%%%

Blue color outlines the original text

Red color indicates the corresponding modifications

GENERAL COMMENT:

The authors of the manuscript entitled ‘Non-orthogonal optical multiplexing empowered by deep learning’ study the non-orthogonal optical multiplexing over a multimode fiber using a deep neural network. They experimentally demonstrate that even input signals sharing the same polarization, wavelength, and spatial position can be explicitly demultiplexed, where the multichannel amplitude and phase-encoded information can be decoded by using only a single-shot speckle intensity. The concept of non-orthogonal optical multiplexing enabled by deep learning is very interesting, which might provide a new way for transmitting information and find potential applications in short-range optical communication and optical encryption. Therefore, I recommend the publication of this manuscript in Nature Communications with the following comments:

OUR REPLY:

We would like to thank the Referee for his/her positive comments “The concept of non-orthogonal optical multiplexing enabled by deep learning is very interesting, which might provide a new way for transmitting information and find potential applications in short-range optical communication and optical encryption” and the support of publication in Nature Communication. We are sorry for our unclear presentation in our previous manuscript. We address his/her important comments point by point in the following, which help us to shape our manuscript.

COMMENT 1:

As I mentioned above, the multichannel light field information can be decoded by using only a single-shot speckle intensity. What is the role of multiple scattering play in non-orthogonal multiplexing? Whether the multiple scattering processes of the multimode fiber enable this kind of decoding?

OUR REPLY:

We thank the Referee for this very important comment. The multiple scattering plays a very important role in non-orthogonal multiplexing based on a single-shot speckle intensity utilizing deep learning. There is residual asymmetry in the optical paths of two multiplexed channels even the phase shift between two channels is compensated. The residual asymmetry mainly comes from imperfect parallelism of two multiplexed beams. As a result, slightly different input wavefronts at the pupil plane of Obj2 will be obtained for the input wavefronts of (A)1&(B)2 and (A+B)1&(0)2 because of the parasitic asymmetry in two optical paths. Here, 1 and 2 indicate the channel number while A and B represent the multiplexed information. As a result, they will excite different sets of modes in the MMF. This difference will be leveraged by the multiple scattering of the MMF, resulting in different output speckle patterns. Therefore, the multiple scattering will result in distinct spatial distribution of speckle intensity even for slightly different non-orthogonal inputs, providing

a distinct identity for a given non-orthogonal input in the parametric space. Therefore, back propagation during the training process can retrieve the mapping function between the non-orthogonal input and its corresponding speckle output.

In order to address this very important comment, we add the following sentence: “**At the same time, the residual asymmetry of the optical paths plays an important role in the non-orthogonal optical multiplexing through the MMF. The residual asymmetry will be leveraged by the multiple scattering of MMF, which can facilitate the solution of multiple-to-one mapping relationship.**”

COMMENT 2:

The fidelity of complex patterns is not as good as the simpler ones. Whether it is limited by the neural network or by the experimental system?

OUR REPLY:

We would like to thank the Referee for this very important comment. Yes, the performance of simpler patterns from the Fashion-MNIST dataset are better than the complex patterns from the CelebA face dataset. We believe that it is limited both by the architecture of the neural network and the amount of training data. The performance of non-orthogonal optical multiplexing can be improved by optimizing the architecture of neural network. More importantly, the demultiplexing fidelity can be further improved and the required amount of data can be reduced by combing a physically-informed model of the multiple scattering system with the neural network [Nat. Commun. 10, 2029 (2019)].

In order to address this important comment, we add the following sentences “**The demultiplexing performance can be further improved by optimizing the network architecture.**” and “**In addition, adding a physically-informed model of the MMF system in the deep neural network could be an effective mean for further boosting the demultiplexing fidelity^{24,25}.**”

COMMENT 3:

The multimode fiber is very sensitive to the perturbation of the environment. How about the robustness of the nonorthogonal optical multiplexing against the perturbation of the environment?

OUR REPLY:

We would like to thank the Referee for this very important comment. Indeed, the environment does affect the stability of the output speckle. We continuously record the output speckle distributions for a fixed multiplexed input and calculate the correlation (Pearson correlation coefficient) between the first snapshot of the output speckle and the corresponding snapshot hereafter. We consider the non-orthogonal multiplexing case of uncorrelated QR codes shown in the Supplementary Figure 3. As can be seen from Fig. R12, the correlation (PCC) between the instantaneous speckle pattern and the first one is varying during the preparation of training and validation datasets. In turn, it indicates that the training of the proposed neural network essentially takes the perturbation from the environment into account, where the neural network shows certain generalization ability against the

perturbation effect from the environment. Therefore, its averaged bit accuracy can be up to 98%.

Fig. R12 The stability monitoring of the MMF under a fixed multiplexed input wavefronts. The PCC evaluating the correlation between an instantaneous speckle pattern (every 100 frames) with the first one is presented.

In order to address this important comment, we add the following sentences in the main text “**At the same time, both the training and validation datasets contain the influence from the environment (see Supplementary Note 4 for more details). According to the retrieved results demonstrated above, the trained SLRnet possesses certain robustness against the perturbation from the environment. More robust demultiplexing can be achieved by using joint training of data collected at different environments^{27,29}.**”

A new Supplementary Note 4 is added. And Fig. R12 is also added in the Supplementary Note 4. “**To monitor the perturbation to the MMF from environment, the PCC evaluating the correlation between an instantaneous speckle pattern (every 100 frames) with the first one is calculated, as shown in Supplementary Figure 5. The non-orthogonal multiplexing case of uncorrelated QR codes shown in the Supplementary Figure 3 is considered. The incident light field is fixed during the measurement. As can be seen from this figure, both the training and validation datasets contain the perturbation from the environment. Therefore, the trained SLRnet possesses certain robustness against the perturbation from the environment.**”

COMMENT 4:

The authors use multimode fiber with different core radii and lengths. Which parameters are more important for achieving high-fidelity decoding?

OUR REPLY:

We would also like to thank the Referee for this important comment. Non-orthogonal optical multiplexing utilizing MMF with different core radii and lengths is used to showcase the generalization of the proposed method for different MMFs. Both parameters are important for realizing non-orthogonal multiplexing. The core radii determine the numbers of mode supported by the MMF. The number of modes should be larger than the number of spatially modulated inputs in

the spatial light modulator for obtaining high-fidelity decoding. While the length of a MMF for a given radius will affect the sensitivity of the MMF to the environment, which will also influence the fidelity of decoding.

In order to address this important comment, we add the following sentence “As can be seen in Figs. 4 and 5, the demultiplexing results of a 1 m MMF is better than the 50 m case. This is because the scattering properties of a longer MMF is much easier to be affected by the environment. The demultiplexing performance can be further improved by optimizing the network architecture.”

COMMENT 5:

Whether the size of the data set will affect the performance of the network?

OUR REPLY:

Yes, the size of the dataset will affect the training process of the network. In order to address this comment, we summary the sizes of all the datasets used in this manuscript and their corresponding retrieved fidelity in the new Supplementary Note 6.

A sentence “The parameters of all used datasets and their corresponding averaged fidelity are summarized in Supplementary Note 6.” is added in the main text.

COMMENT 6:

Why they chose the wavelength of 532 nm for the experiment? It is not the standard communication wavelength.

OUR REPLY:

We would also like to thank the Referee for this comment which helps us to avoid confusion. Choosing the wavelength of 532 nm for the experiment is served as a proof-of-concept demonstration of non-orthogonal multiplexing. The demonstrated concept can be generalized to other working wavelengths, if the spatial light modulator and CMOS camera are replaced by the ones whose working spectrum range are fitted. At the same time, the diameter of the MMF should be large enough for supporting sufficient number of guided modes at this new wavelength.

In order to address this comment, we change the following sentence “A monochromatic laser with a power of 50 mW (\$\lambda = 532\$ nm, MSL-S-532 CH80136, CNI) is used as the light source (see Supplementary Note 4 for more details regarding to the experimental setup).” to “A monochromatic laser with a power of 50 mW (\$\lambda = 532\$ nm, MSL-S-532 CH80136, CNI) is used as the light source (see Supplementary Note 4 for more details regarding to the experimental setup), where it can be generalized to other wavelengths.”

In summary, we would like to thank these very important comments from the Referee which helped us to substantially improved the quality of our manuscript. We hope that the content of our revised manuscript is now suitable for Nature Communications. Your consideration is greatly appreciated.

REVIEWER COMMENTS

Reviewer #1 (Remarks to the Author):

This version of the paper is significantly improved in many of the machine learning aspects, and it is good to see more challenging datasets used successfully.

I think there is still room for improvement of the discussion around the approximation to the inverse ('Comment 1' for Reviewer 1 in the rebuttal document).

The demonstration of the multiplexed transmission of QR codes that are consisted of uncorrelated digits 0 and 1 first. Is an interesting result, as random data provides the ultimate test for the quality of the approximate inverse. However it is only a very low 20x20 resolution. I think you would get to the core of the problem of the approximation of an inverse matrix if you show a graph of performance at a series of increasing resolutions for the random input matrices (e.g. 20x20, 50x50, 100x100, up to the max 200x200 which you used in training) as I suspect your performance will tail off significantly as you get larger matrices to approximate, highlighting the core challenge here. For images we can use the benefits of ML to use smoothness in images to find a compact embedding, but that can't be done for random pixel data.

I'm not sure it is helpful to call these 'QR codes' unless you are actually using a standard QR code format and were e.g. checking to see whether you would still get the correct code. It seems like you are just using random data, so I would remove the QR code term.

For 'Comment 4' of the rebuttal can you confirm that the 'QR results' are on test data, not training data, and add that to the paper?

So in summary:

- improve detail on approximate inverse
- extend random results to show how quickly performance decays with higher resolution as this will clearly show the limitations of the approach (exponentially increasing requirements in training data and network size as resolution increases).

Reviewer #2 (Remarks to the Author):

This manuscript has been revised accordingly, and the current version can be accepted for publication in principle.

Reviewer #3 (Remarks to the Author):

The authors have answered extensively to all the reviewers' comments with supplemented experimental results and network optimizations. Considering the other reviewer's comments and how the authors responded, I can recommend the acceptance of the current manuscript.

Just one minor suggestion: I find that there still exist some typos in the main text, and the authors are suggested to check the manuscript carefully.

Reviewer #4 (Remarks to the Author):

In my opinion, the authors have addressed successfully all of the comments/suggestions raised by all the reviewers, and the present manuscript is ready for acceptance.

%%%%%%%%%

Response to the Referee 1

%%%%%%%%%

Blue color outlines the original text,

Red color indicates the corresponding modifications

GENERAL COMMENT:

This version of the paper is significantly improved in many of the machine learning aspects, and it is good to see more challenging datasets used successfully. I think there is still room for improvement of the discussion around the approximation to the inverse ('Comment 1' for Reviewer 1 in the rebuttal document).

OUR REPLY:

We would like to thank the Referee again for his/her inspired comments which substantially help us to improve the quality of our manuscript. We really appreciate that the Referee recognized the improvement of our revised manuscript. We will address his/her comments in the following.

In order to improve the discussion around the approximation to the inverse, we add the following sentences "This neural network builds an approximate relationship that maps the speckle intensity at the distal end of the MMF to the distributions of amplitude and phase for several input light fields at the proximal end of the MMF, where the training of the network relies on the dataset using pairs of output speckles and their corresponding input wavefronts." in the main text.

COMMENT 1:

The demonstration of the multiplexed transmission of QR codes that are consisted of uncorrelated digits 0 and 1 first. Is an interesting result, as random data provides the ultimate test for the quality of the approximate inverse. However it is only a very low 20x20 resolution. I think you would get to the core of the problem of the approximation of an inverse matrix if you show a graph of performance at a series of increasing resolutions for the random input matrices (e.g. 20x20, 50x50, 100x100, up to the max 200x200 which you used in training) as I suspect your performance will tail off significantly as you get larger matrices to approximate, highlighting the core challenge here. For images we can use the benefits of ML to use smoothness in images to find a compact embedding, but that can't be done for random pixel data.

OUR REPLY:

We really thank the Referee for this important comment. Yes, the referee is right. Following the suggestion of the Referee, we perform the experiments for the multiplexing cases of 20×20 , 40×40 , 50×50 , 100×100 and 200×200 , where the bit accuracy rate is decreased when the resolution is increased. The sizes of all datasets are fixed at 30,000 pairs. The results are shown in Fig. R1, where two amplitude and phase encoded wavefronts are used for multiplexing. As can be seen from this figure, there is still a challenge in the proposed approach when the capacity of the input information is enlarged, where increasing the resolution of input multiplexing channels will require exponentially increasing amounts of data.

Fig. R1 The dependence of bit accuracy rate on the input pixel number. The total input pixel number can be calculated by resolutions $\times 2$ (amplitude and phase) $\times 2$ (beam number), as indicated in the inset.

As pointed out the Referee, physically-informed approaches could be an effective mean to solve this challenge. In order to address this comment, we change the sentence “In addition, adding a physically-informed model of the MMF system in the deep neural network could be an effective mean for further boosting the demultiplexing fidelity^{24, 25}.” to “There are still challenges to overcome in this data-driven approach. A typical one is the ability to multiplex information with higher capacity will require exponentially increasing amounts of data. Adding a physically-informed model of the MMF system in the deep neural network could be an effective solution for this challenge, which would also further boosting the demultiplexing fidelity^{24, 25}.”

COMMENT 2:

I’m not sure it is helpful to call these ‘QR codes’ unless you are actually using a standard QR code format and were e.g. checking to see whether you would still get the correct code. It seems like you are just using random data, so I would remove the QR code term.

OUR REPLY:

We thank the Referee for this important comment. Yes, the Referee is right. In order to address this comment, we change the term “QR codes” to “random binary data” throughout our manuscript.

COMMENT 3

For ‘Comment 4’ of the rebuttal can you confirm that the ‘QR results’ are on test data, not training data, and add that to the paper?

OUR REPLY:

We confirm these results of the ‘QR results’ are on the test set. In order to address this comment, we add a sentence “These results are on the test set.” in the caption of Fig. S3.

COMMENT 4

So in summary:

- improve detail on approximate inverse
- extend random results to show how quickly performance decays with higher resolution as this will clearly show the limitations of the approach (exponentially increasing requirements in training data and network size as resolution increases).

OUR REPLY:

In summary, we really thank the Referee for these important comments which substantially help us to shape our manuscript. It is anticipated that the revised manuscript is now can meet the criteria of Nature Communication and your consideration is sincerely appreciated.

%%%%%%%%%

Response to the Referee 2

%%%%%%%%%

GENERAL COMMENT:

This manuscript has been revised accordingly, and the current version can be accepted for publication in principle.

OUR REPLY:

We really appreciate the Referee for his/her important comments which substantially help us to improve the quality of our manuscript.

%%%%%%%%%

Response to the Referee 3

%%%%%%%%%

GENERAL COMMENT:

The authors have answered extensively to all the reviewers' comments with supplemented experimental results and network optimizations. Considering the other reviewer's comments and how the authors responded, I can recommend the acceptance of the current manuscript.

OUR REPLY:

We really thank the Referee for his/her important comments which substantially help us to improve the quality of our manuscript.

COMMENT 1:

Just one minor suggestion: I find that there still exist some typos in the main text, and the authors are suggested to check the manuscript carefully.

OUR REPLY:

We thank the Referee for this suggestion. We carefully check the manuscript again, where the revised typos are outlined in red in the marked PDF file.

%%%%%%%%%

Response to the Referee 4

%%%%%%%%%

GENERAL COMMENT:

In my opinion, the authors have addressed successfully all of the comments/suggestions raised by all the reviewers, and the present manuscript is ready for acceptance.

OUR REPLY:

We really appreciate the Referee for his/her important comments which substantially help us to improve the quality of our manuscript.

REVIEWERS' COMMENTS

Reviewer #1 (Remarks to the Author):

I had a quick look. I think if the authors include Figure R.1. from the rebuttal in the supplementary material beside the random binary analysis in Figure 2 with a brief description of the experiment, then I think the paper is ready, as this highlights the strengths and weaknesses of the approach clearly.

%%

Response to the Referee 1

%%

Blue color outlines the original text,

Red color indicates the corresponding modifications

GENERAL COMMENT:

I had a quick look. I think if the authors include Figure R.1. from the rebuttal in the supplementary material beside the random binary analysis in Figure 2 with a brief description of the experiment, then I think the paper is ready, as this highlights the strengths and weaknesses of the approach clearly.

OUR REPLY:

We would like to thank the Referee again for his/her important comments which substantially help us to improve the quality of our manuscript. We address his/her comment in the following:

In order to address this comment, we follow the suggestion to modify the figure in the Supplementary material. The revised figure with the adding part of Fig. R1 (b) is shown by Fig. R1. We revise the sentence “A typical one is the ability to multiplex information with higher capacity will require exponentially increasing amounts of data.” to “A typical one is the ability to multiplex information with higher capacity will require exponentially increasing amounts of data (see Supplementary Note 2).” in the main text.

We also add the following sentences “It should be pointed out that when the multiplexing capacity is increased, the bit accuracy rate is decreased with a fix amount of dataset, as shown by Supplementary Figure 3b. It indicates a challenge of the proposed approach, where increasing the resolution of input multiplexing channels will require exponentially increasing amounts of data.” and “b The dependence of bit accuracy rate on the input pixel number. The total input pixel number can be calculated by resolutions $\times 2$ (amplitude and phase) $\times 2$ (beam number), as indicated in the inset. The sizes of all datasets are fixed at 30,000 pairs.” in the Supplementary Note 2.

Fig. R1 Demultiplexing results for the non-orthogonal optical multiplexing of binary random binary data in a 1m MMF. a The speckle outputs, the corresponding ground truths, the retrieved results of neural network, the binarized results and the bit accuracy rate are given, respectively. The length of the MMF is 1m. It should be pointed out that the SLRnet here is revised to facilitate the demultiplexing of the random binary data, as elaborated in our new Supplementary Note 7. The amplitude and phase are self-normalized. The ground truths of uncorrelated random binary data are generated by the uniform

random function of Matlab, which consists of 20×20 pixels. These results are on the test set. **b** The dependence of bit accuracy rate on the input pixel number. The total input pixel number can be calculated by $\text{resolutions} \times 2(\text{amplitude and phase}) \times 2$ (beam number), as indicated in the inset. The sizes of all datasets are fixed at 30,000 pairs.

In summary, we really thank the Referee for his/her important and inspired comments which substantially help us to shape our manuscript. It is anticipated that the revised manuscript is now can meet the criteria of Nature Communication and your consideration is sincerely appreciated.